# Abstract rule learning promotes cognitive flexibility in complex environments across species

Florian Bähner [1,2] ✉, Tzvetan Popov[3,4], Nico Boehme[1,2], Selina Hermann[1,2], Tom Merten [1,2], Hélène Zingone [1,2], Georgia Koppe [2,5,6], Andreas Meyer-Lindenberg[2,12], Hazem Toutounji [7,8,9,12] & Daniel Durstewitz [6,10,11,12]

Rapid learning in complex and changing environments is a hallmark of intelligent behavior. Humans achieve this in part through abstract concepts applicable to multiple, related situations. It is unclear, however, whether the computational mechanisms underlying rapid learning are unique to humans or also exist in other species. We combined behavioral, computational and electrophysiological analyses of a multidimensional rule-learning paradigm in male rats and in humans. We report that both species infer task rules by sequentially testing different hypotheses, rather than learning the correct action for all possible cue combinations. Neural substrates of hypothetical rules were detected in prefrontal network activity of both species. This species-conserved mechanism reduces task dimensionality and explains key experimental observations: sudden behavioral transitions and facilitated learning after prior experience. Our findings help to narrow the explanatory gap between human macroscopic and rodent microcircuit levels and provide a foundation for the translational investigation of impaired cognitive flexibility.

Cognitive flexibility is critical for the ability to respond to changes in the environment in adaptive ways. Deficits in this domain are observed in several major neuropsychiatric disorders[1]. However, it is not fully understood how correct behavioral rules are identified in ever-changing contexts and how this information is encoded in neural activity[2–7]. A powerful theoretical framework in this area is reinforcement learning (RL), which describes how action values in different environmental situations (known as states) are learned to guide decisions and maximize reward[8]. This framework has gained support from the neurosciences following the discovery of neural substrates of RL quantities such as action values (i.e., the expected reward from taking that action) and reward prediction errors (i.e., the difference between actual and expected reward)[9,10]. A core assumption of classic RL models is that subjects learn to map each state of the environment to the maximum-value action[8]. However, the real world is multi-dimensional, where many sources of information, such as sensory

[1]RG Behavioral Physiology in Psychiatry, Central Institute of Mental Health, Medical Faculty Mannheim, Heidelberg University, Mannheim, Germany. [2]Department of Psychiatry and Psychotherapy, Central Institute of Mental Health, Medical Faculty Mannheim, Heidelberg University, Mannheim, Germany. [3]Department of Psychology, University of Zurich, Zurich, Switzerland. [4]Department of Psychology, University of Konstanz, Konstanz, Germany. [5]Hector Institute for Artificial Intelligence in Psychiatry, Central Institute of Mental Health, Medical Faculty Mannheim, Heidelberg University, Mannheim, Germany. [6]Interdisciplinary Center for Scientific Computing, Heidelberg University, Heidelberg, Germany. [7]School of Psychology, University of Sheffield, Sheffield, UK. [8]Insigneo Institute for in silico Medicine, University of Sheffield, Sheffield, UK. [9]The Neuroscience Institute, University of Sheffield, Sheffield, UK. [10]Department of Theoretical Neuroscience, Central Institute of Mental Health, Medical Faculty Mannheim, Heidelberg University, Mannheim, Germany. [11]Faculty of Physics and Astronomy, Heidelberg University, Heidelberg, Germany. [12]These authors contributed equally: Andreas Meyer-Lindenberg, Hazem Toutounji, Daniel Durstewitz. ✉e-mail: florian.baehner@zi-mannheim.de

cues, reward history, and working memory of past choices, combine to form each state. This results in an exponential growth in the number of states and state-action mappings (known as the curse of dimensionality) that translates into slow and gradual learning[11]. Consequently, standard RL models struggle to explain hallmarks of flexible behavior such as sudden transitions in performance[7,12–15], rapid learning in complex environments, or faster learning with prior experience[3,11,16].

A long tradition in cognitive science maintains that humans work around the curse of dimensionality by learning abstract concepts like rules[3,17], categories[18,19], or schemas[20,21] that can be applied to multiple, related situations. Recent advances in artificial intelligence inspired by cognitive neuroscience have suggested different computational mechanisms for learning abstract models of the world that facilitate generalization of knowledge and transfer of learned skills to new tasks[3,16,22–26].

Since one-to-one mappings of states to actions are inefficient in high-dimensional environments with many potential task contingencies, subjects may rather learn and test *hypothetical rules* underlying these mappings. These hypothetical rules summarize different combinations of environmental cues, actions, and potential outcomes within a common concept, such as *don't press any lever paired with a bright light, regardless of lever location*. This abstraction reduces dimensionality dramatically since every cue becomes an instance of a general task feature, allowing the learner to direct its attention toward this feature and ignore other cues. The learner can then test different hypothetical rules against environmental evidence until they successfully identify the experimenter-defined rule. We call such hypothetical rules *behavioral strategies* in the following to distinguish them from task rules.

Indeed, flexible human behavior can be explained by computational models that are compatible with this framework, including hierarchical and attention-modulated RL models[3,5,16,27]. It is unknown, however, how rodents infer task rules in complex environments and whether some of the underlying behavioral and neural mechanisms are conserved across species. Pioneering work has shown that in some contexts, rats do follow behavioral strategies like *win-stay-lose-shift* or *spontaneous alternation*[28,29], and recent methodological work has shown that such strategies can be identified in rodents, macaques, and humans[30]. However, these findings have not been generalized to cognitive flexibility or empirically contrasted to the prevailing view of classical RL models.

We developed a novel translational rule-learning task and used a combination of behavioral, computational, and electrophysiological methods to test whether common computational mechanisms exist in both species. We found that both rats and humans followed low-dimensional behavioral strategies to test each task feature for relevance rather than learning high-dimensional mappings between environmental states and actions. Moreover, we decoded strategy-related quantities with features of abstract, hypothetical rules from both rat multiple-single unit and human magnetoencephalography (MEG) recordings in the prefrontal cortex (PFC), highlighting a common mechanism of flexible rule learning.

## Results

### Rats infer task rules using low-dimensional strategies

We developed a novel dual-choice multidimensional rule-learning paradigm with four relevant task features to evaluate the hypothesis that rats identify experimenter-defined rules by sequentially testing different low-dimensional behavioral strategies (Fig. 1a). A loudspeaker and a cue light were positioned above each lever. In each trial, one auditory and one visual cue were presented to model multisensory input. Rats received lever training and were exposed to task cues, but were not pre-trained on any rule. They had to learn one out of eight possible task rules with deterministic reward feedback: go-click (i.e., follow the auditory cue in the presence of a distracting visual cue), go-silent, go-light, go-dark, go-right, go-left, alternate, or win-stay-lose-shift/WSLS. Thus, four task features could be relevant for reward in every trial: the visual and auditory cues, as well as outcome and choice histories. Rats were able to learn each rule, albeit at different learning rates (Supplementary Table 4). We used change point analysis and behavioral modeling to identify when performance increases occurred during rule learning and how abrupt they were (Supplementary Methods)[31,32]. We found that sudden transitions in performance (median ($Q_1$–$Q_3$) of 10–90% rise time: 1 (1-1) trials, $N = 117$; e.g., Fig. 1d) are similar to what has been observed in other, simpler learning paradigms[7,12,13]. This finding is at odds with the classic RL view that assumes gradual learning because rats would need to figure out the correct action (e.g., right or left lever press) for each of $2^4 = 16$ states if they learned the full state-action space. We therefore propose an alternative hypothesis (Fig. 1b) where rats use up to eight low-dimensional behavioral strategies (*go-light/go-dark, go-click/go-silent, go-right/go-left, alternate, or win-stay-lose-shift*) to sample different features for relevance (i.e., significance of sensory cues, past choices, and outcome history). In this scenario, rats would test different simple hypothetical rules sequentially until they infer the correct task rule.

To test this, we developed a strategy detection algorithm that assesses whether a particular strategy is more likely than chance or than any other strategy within every possible trial window (Methods, Fig. 1b). The algorithm identifies trial sequences within each session in which a particular strategy is the winning strategy. In each task rule, animals indeed followed a median of 7 (6–8) out of 8 possible strategies (Supplementary Table 4). The average length of an individual strategy sequence (i.e., the number of consecutive trials where a strategy was followed) was 14 (12–16.3) trials. Interestingly, we replicated this finding in a separate cohort of rats ($N = 19$) that were presented with random reward feedback (i.e., none of the four task features were predictive of reward), indicating that hypothesis testing reflects a general cognitive strategy, even when there is no behavioral advantage.

We conducted several analyses to support our hypothesis (Methods). First, we applied our detection algorithm on synthetic strategy sequences to show that it has the required sensitivity to detect strategies from data where the ground truth is known (Fig. 1c, bottom). Moreover, we show that it is highly unlikely that the detected strategies were false positives, since the percentage of trials for which a strategy could be detected was significantly higher in naïve rats ($N = 159$ sessions) compared to a simulated agent making random choices (10,000 sessions; Fig. 1c top). Second, we excluded the possibility that rats were using more complex, higher-dimensional strategies such as *win stay-lose shift visual*, where the action in response to the current visual cue also depends on prior reward. We assessed whether rats use a *win stay-lose shift* vs. *win shift-lose stay* approach with respect to place, visual, and auditory cues. However, our detection algorithm revealed no statistical difference in detecting these six higher-dimensional strategies between rats and the randomly permutated synthetic data ($p = 0.55$, two-sided Mann–Whitney test; Fig. 1c top). Moreover, the percentage of trials for which a strategy could be detected was significantly higher for low- compared to high-dimensional strategies in rats ($p = 1.46 \times 10^{-27}$, two-sided Wilcoxon signed rank test). Third, we tested whether the observed pattern of low-dimensional strategies can predict learning. To achieve this, we defined an empirical measure for learning based on the comparison between the length of a correct strategy sequence (i.e., the strategy that corresponds to the current task rule) and the distribution of sequence lengths of that same strategy as detected in the cohort that received random reward. For example, we assume a rat learned the rule go-silent successfully when the sequence length of the corresponding *go-silent* strategy was at least 21 trials. Beyond this threshold, *go-silent* sequences were considered outliers in the random reward cohort (Supplementary Methods, Supplementary Table 4). The trial at which rats reached the empirical learning criterion correlated

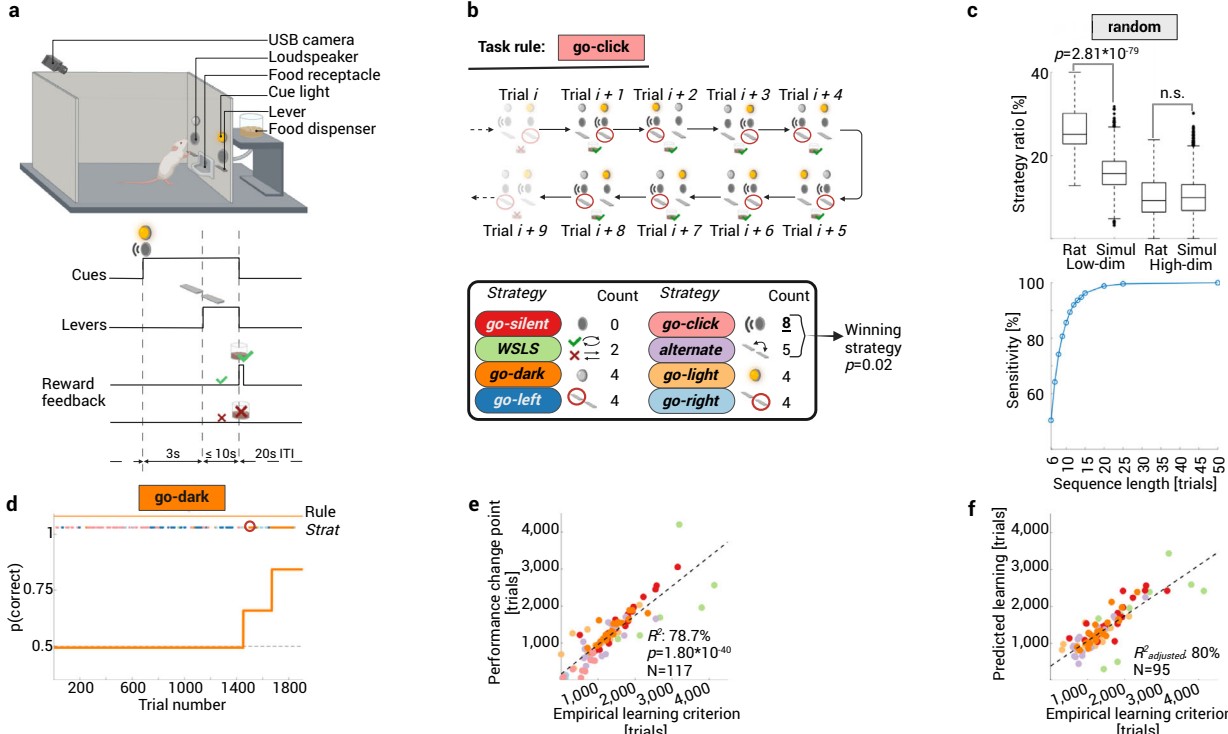

**Fig. 1 | Rats test low-dimensional strategies to infer the task rule. a** Each trial starts with the presentation of one auditory and one visual cue. Three seconds later, two levers extend and rats use trial-and-error to learn the correct choice. Depending on the experimental rule, one out of four task features can be relevant for reward. ITI/inter-trial interval. **b** *Top:* example of a *go-click* sequence consisting of eight consecutive trials as detected with the strategy detection algorithm during the task rule go-click (rectangular shape). For each trial, the current sensory input is shown as well as the choice and reward feedback. *Bottom:* count of trials that are consistent with each strategy (oval-shaped icon). The binomial statistic (Methods) based on this count indicates that the rat followed the strategy *go-click*. **c** Validation of the strategy detection algorithm. Top: higher strategy ratio (i.e., the percentage of trials for which a strategy could be detected) for eight low-dimensional strategies in naive rats performing the random rule vs. a random agent (two-sided Mann–Whitney test). In contrast, the strategy ratio for six high-dimensional strategies was not higher than chance (two-sided Mann–Whitney test). There was a significant difference between the ratios for low- vs. high-dimensional strategies ($p = 1.46 \times 10^{-27}$, two-sided Wilcoxon signed rank test). Box plots showing median, 25%–75% percentile, whiskers: 1.5 IQR, and outliers. *Bottom:* sensitivity of the algorithm was determined using synthetic data (Methods) and depends on sequence length (statistical power >80% for sequence lengths ≥9 trials). **d** Learning curve of an individual rat. Task rule go-dark (upper solid line) and detected strategy sequences (lower lines) are color-coded, the learning curve is visualized using a sigmoidal model fit to the data, with the steepness of the slope indicating the speed of transition in choice behavior (Supplementary Methods). The empirical learning criterion is marked with a circle. **e** High correlation between learning criterion and performance change point, each dot is a rat learning one out of seven different rules (Pearson correlation). **f** A linear regression model shows that several indicators of strategy usage significantly predicted the learning trial ($R^2_{adjusted}$). See Supplementary Tables 5 and 6. Created in BioRender. Böhme, N. (2025) https://BioRender.com/k38n620. Source data are provided as a Source Data file.

---

significantly with abrupt performance increases (Fig. 1d, e). Furthermore, we reasoned that if rats follow strategies to identify task rules, more efficient strategy use should lead to faster learning. For instance, we could expect that faster learners do not revisit poorly performing strategies and do not abandon a successful strategy after they discover it leads to reward. Indeed, multiple regression analysis revealed that several strategy-associated quantities were predictive of the trial at which the empirical learning criterion was reached ($R^2_{adjusted} = 80\%$; Fig. 1f). These included the trial number at which the animal followed the correct strategy for the first time ($p = 1.6 \times 10^{-4}$), the number of correct strategy sequences before reaching criterion ($p = 5.0 \times 10^{-10}$), a perseveration measure[33] indicating that poorly performing strategies are revisited ($p = 9.0 \times 10^{-20}$), and a categorical variable indicating the target rule (Supplementary Tables 5 and 6). While a number of trials (36.16 ± 1.24%) could not be assigned to a specific strategy, this number was not predictive of learning since adding it to the regression model did not increase goodness of fit ($R^2_{adjusted} = 78.9\%$, $p = 0.43$ for this regressor). We found qualitatively comparable results in a classic operant two-rule set-shifting task[33] with less task features, where rats ($N = 16$) also followed behavioral strategies that predicted sudden transitions in performance (Supplementary Methods, Supplementary Fig. 1a, b). However, rats reached criterion earlier in this task, which is

in line with the idea that learning is faster because there were fewer strategies to explore (Supplementary Fig. 1c). In sum, these analyses provide a first indication that rats test low-dimensional behavioral strategies to infer task rules rather than learning high-dimensional mappings between environmental states and actions.

## Strategy-specific attention reduces task dimensionality

We next aimed to identify computational processes that lead to dimensionality reduction during strategy-based learning. Selective attention is a candidate mechanism because it allows animals to focus on one task feature at a time, rather than dividing attention between low-level cues (Fig. 2a). This has been shown to result in more efficient learning in complex environments[5,11]. Attention thus acts like a filter such that only part of the state space (Fig. 2a) is relevant for choice and learning when individuals test different hypothetical rules (e.g., *go-click*). This sub-space corresponds to the *current attention focus* (e.g., the task feature auditory cue). Strategies thus map to distinct state spaces via their attention focus.

To test the role of attention in our paradigm, we applied machine learning techniques (Methods) to screen videotaped sessions for behavioral markers of attention in rats performing multiple consecutive rule switches ($N = 29$ rats). We found a strong orientation

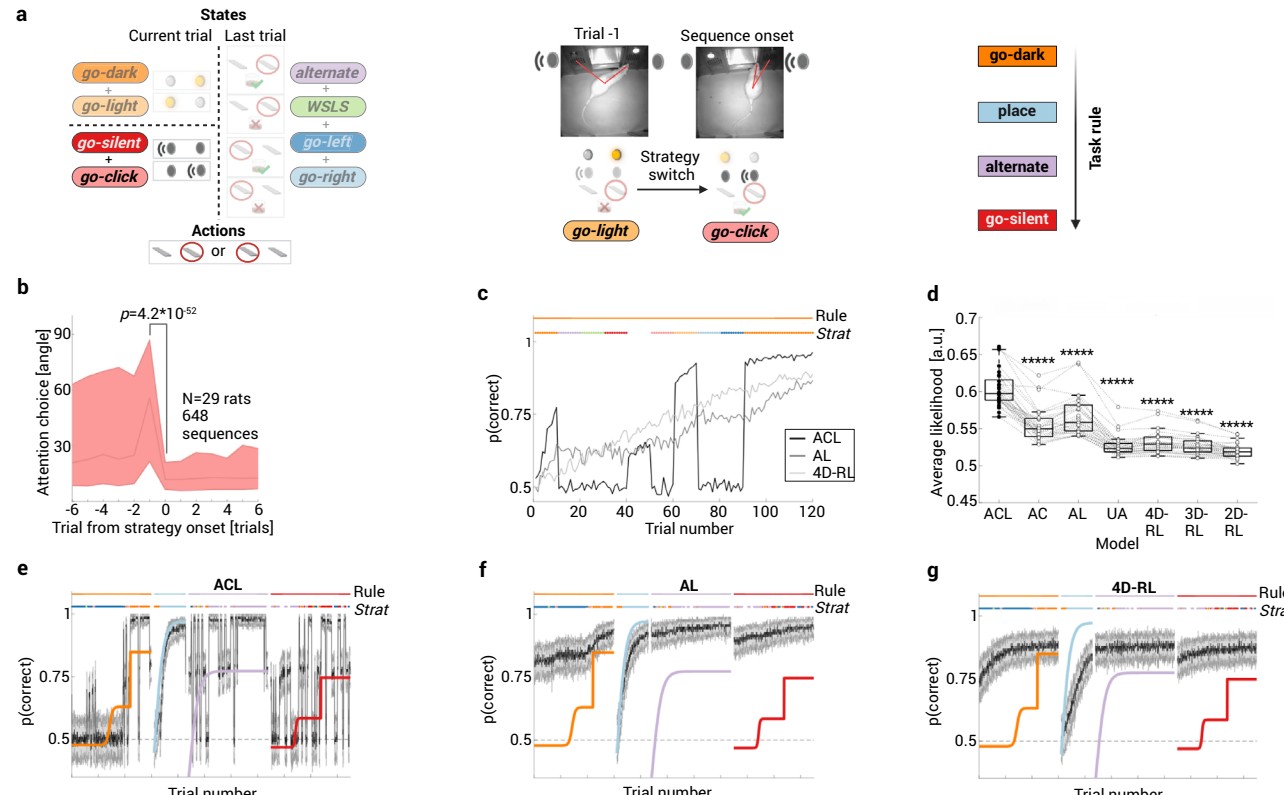

**Fig. 2 | Strategy-specific attention reduces task dimensionality. a** *Left:* s-a space used in attention-based RL models. Attention acts like a filter such that only part of the state space is relevant (*current attention focus*). Strategies map to distinct state spaces via their attention focus. *Right:* attention is quantified using head angles while rats performed multiple, consecutive rule switches. Example of an attentional shift from light to auditory cue with a corresponding decrease in head angle. **b** Head-direction plot for strategy *go-click* during choice formation. Head angles with respect to sound are aligned to cue onset and abruptly decrease at strategy onset (trial 0). Data presented as median, first and third quartile, two-sided Wilcoxon matched pairs test. See Supplementary Figs. 2–4. **c** Example learning curves of RL agents (gray scale lines) trained on the task rule go-dark (upper solid orange line), averaged over 1000 simulations (learning rate $\alpha = 0.2$; exploration rate $\beta = 3$). Agents switch attention between different strategies, before settling on the correct strategy (lower color-coded lines). Only models with an attention-at-choice component had step changes. **d** Trial-averaged cross-validated likelihoods for seven RL model variants (*N* = 29 rats). Box plots showing median, 25%–75% percentile,

whiskers: 1.5 IQR and outliers; dashed lines connect values (circles) of the same rat; filled circles represent best (i.e., highest likelihood) model for each rat. The ACL model, where attention biases both choice and learning, was the winning model in each rat, which indicates that attention acts like a strong filter that reduces task dimensionality. Model comparisons between ACL and all other models are Benjamini–Hochberg corrected (two-sided Wilcoxon matched pairs test with *****$p < 10^{-5}$). **e**–**g** Representative example of model simulations using model parameters estimated from experimental data of one rat. Color-coded sigmoidal learning curves based on choice behavior show transitions instead of gradual learning. Learning curves were truncated and only show the part where changes occurred (spacing between ticks on the *x*-axis is 200 trials). Sudden performance changes were only reproduced if choice is modulated by strategy-specific attention (ACL in **e**), but not AL (**f**) or 4D-RL models (**g**). Created in BioRender. Böhme, N. (2025) https://BioRender.com/a62a760. Source data are provided as a Source Data file.

reaction at cue onset that was influenced by the current strategy. More specifically, we found that the onset and offset trials of each strategy sequence were locked to abrupt changes in head direction at cue onset towards and away from the task feature that is relevant for that strategy (Fig. 2b). This allowed us to obtain a trial-by-trial measure of the rat's attention to each task feature at the time of cue presentation, which we term attention-at-choice. We detected strategy-specific abrupt changes in attention to the corresponding task feature for all eight strategies (Supplementary Fig. 2, Supplementary Table 1). A similar, but feature-independent, orientation reaction occurred during reward feedback, which we term attention-at-reward (Methods). We replicated these findings in two cohorts of rats receiving random reward (Supplementary Figs. 3 and 4, Supplementary Tables 2 and 3). This adds to the evidence that, in the context of random reinforcement, choices are not random (Fig. 1c) but correspond to goal-directed actions that can be predicted from head movements preceding choice. Rats thus test different hypotheses sequentially, even when they have no incentive to follow a specific strategy. Based on head-direction plots, it is difficult to exclude that rats focus their attention on more than one task feature. However, we also show in three different

experimental contexts that some strategies (*alternate, go-left, go-right*) can already be identified based on movement patterns in the inter-trial interval (ITI) preceding each trial (Supplementary Fig. 5). This indicates that at least in some cases, rats exclusively focus on a single task feature.

To assess whether attention leads to dimensionality reduction during rule learning, we incorporated binarized (based on the currently detected strategy) and continuous (based on head-movement patterns) attention measures into different RL models (Methods). We evaluated how rats learn rules in the state-action/s-a space defined by the four task features (where the state space consists of three subspaces: a visual, an auditory, and a 2-dimensional history state combining the place and outcome features; Fig. 2a). More specifically, we tested whether attention modulates behavior during choice alone (attention-at-choice/AC model), learning (credit assignment during value update) alone (attention-at-learning/AL model), or both (ACL model)[11]. The attention measures bias learning and choice in the ACL model such that one task feature has a privileged position: the 4-dimensional state space of the task collapses into the 1-dimensional sub-space related to the current strategy. In a fourth model, attention

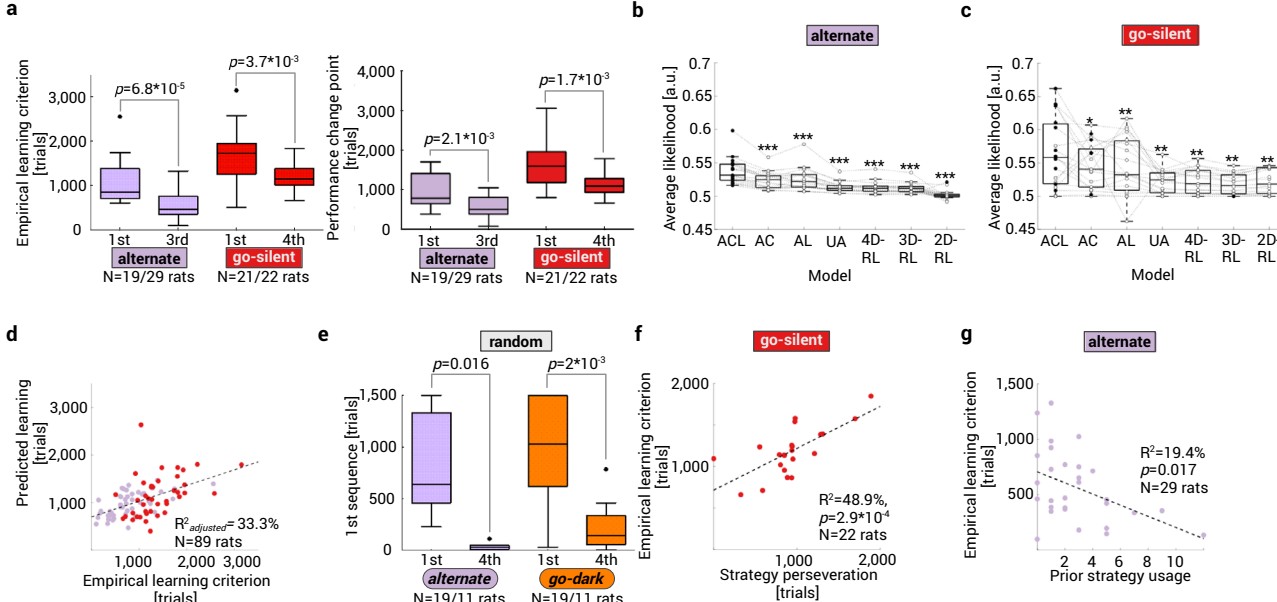

**Fig. 3 | Dimensionality reduction also contributes to transfer effects. a** Faster rule learning with prior experience. Both the learning trial (*left*) and the performance change point (*right*) occurred earlier in experienced rats learning the task rules alternate/go-silent as compared to naïve animals (two-sided Mann–Whitney tests). Replication of RL model findings (as in Fig. 2d) for rats learning alternate (*N* = 19; **b**) and go-silent (*N* = 19; **c**) as a first rule. Dashed lines connect values (circles) of the same rat; filled circles represent the best (i.e., highest cross-validated likelihood) model of each rat. Model comparisons between ACL and all other models are Benjamini–Hochberg corrected two-sided Wilcoxon matched pairs test with *p < 0.05, **p < 0.01, ***p < 10⁻³. See Supplementary Fig. 6b, c for RL models with continuous attention scores. **d** Scatter plot showing results from the regression model using data from naïve and experienced rats learning the rules alternating and go-silent. The learning trial was predicted ($R^2_{adjusted}$) by median

attention-at-choice ($p = 1.6 \times 10^{-5}$) but not median attention-at-reward ($p = 0.86$). A variable coding for learning experience was also significant ($p = 4 \times 10^{-3}$). Each dot represents the empirical vs. predicted learning trial. **e** In the absence of a task rule, strategies are selected earlier if they have been previously reinforced (i.e., in experienced rats/fourth rule) as compared to a naïve cohort (first rule), two-sided Mann–Whitney test. **f** Perseveration on the previously reinforced strategy *alternate* predicts when the rule go-silent is learned, indicating a negative transfer effect (each dot corresponds to one rat, Pearson correlation). **g** The number of times a rat used the strategy *alternate* in previous rules predicted how fast a rat would learn the task rule alternate (Pearson correlation). Box plots showing median, 25%–75% percentile, whiskers: 1.5 IQR, and outliers. Created in BioRender. Böhme, N. (2025) https://BioRender.com/b41u710. Source data are provided as a Source Data file.

is split equally between all three subspaces so that it neither biases choice nor learning (uniform-attention/UA model). We also compared these attention-modulated RL models to three variants of a standard RL model that mapped a multidimensional state space directly to actions. These standard models differed in which task features comprised the relevant states: (1) a 4-dimensional RL model (4D-RL) where the state space consisted of all four relevant task features, resulting in 32 s-a mappings to be learned simultaneously, (2) a 3-dimensional RL model (3D-RL) that excluded the outcome feature, and (3) a 2-dimensional RL model (2D-RL) that only included the sensory task features.

We first compared the models through simulations of artificial agents that aimed to learn an experimenter-defined rule (go-dark in this example; Fig. 2c). We found that attention during choice (ACL and AC models) is necessary to produce sudden transitions in the agent's behavior as it learned the correct task rule. In contrast, models where attention-modulated learning alone (AL model) or where the agent learned all or partial mappings between states and actions (UA model and standard RL models) resulted in slow and gradual learning.

Next, we fitted the parameters of the four attention-modulated RL models and the three standard RL models to experimental data, followed by cross-validated model comparison. This revealed that the ACL model was indeed the best at predicting held-out behavioral data in all 29 rats. Most importantly, there was no statistical difference between the binary ACL model (Fig. 2d) that assumes the rat is attending fully to the current cue and the continuous ACL model based on measured head angles ($p = 0.87$, two-sided Wilcoxon signed rank test; Supplementary Fig. 6a). This indicates that attention is a very strong filter and is thus additional evidence for low-dimensional

strategies. We further confirmed that attention-at-choice is necessary to capture sudden improvements in performance, a core feature of rat multidimensional rule learning: we ran model simulations using model parameters as estimated from the experimental data and constrained by sensory cues and attention scores as measured experimentally. We found that sudden performance changes could only be reproduced if choice is modulated by strategy-specific attention (Fig. 2e–g). To further test whether all strategies are necessary to predict held-out behavioral responses or whether subsets of them suffice, we compared the binary ACL model to three model variants where one of the s-a spaces (either auditory, visual, or place-outcome) was treated as in the UA model. This corresponds to a scenario in which strategy-specific attention effects for either auditory (*go-click*, *go-silent*), visual (*go-light*, *go-dark*), or place-outcome strategies (*alternate*, *go-left*, *go-right*, and *win-stay-lose-shift*) are removed. Removal of attention effects related to any of the s-a-spaces resulted in significantly worse prediction of held-out behavioral responses (Supplementary Fig. 6d).

In summary, our data suggest that selective attention organizes action selection and learning when individuals test different hypothetical rules and that this computational mechanism speeds up learning by reducing task dimensionality in complex environments.

## Dimensionality reduction also contributes to transfer effects

Prior experience often leads to faster learning in similar tasks[3,11,16]. We also observed that experienced rats acquired the task rules alternately and go-silent (as a third and fourth rule) faster than naïve animals (Fig. 3a). Below, we tested whether dimensionality reduction during strategy-based learning contributes to such transfer effects in two different ways.

Given that attentional mechanisms were similar for both naïve (Fig. 3b, c, Supplementary Fig. 6b, c) and experienced rats (Fig. 2d), we first tested whether faster learning in experienced rats is related to more focused attention. More specifically, we estimated a regression model ($R^2_{adjusted} = 33.3$ %; Fig. 3d) and found that the learning trial can be predicted by median attention-at-choice (for all trials preceding the learning trial; $p = 1.6 \times 10^{-5}$) but not median attention-at-reward ($p = 0.86$). A variable coding for learning experience was also significant ($p = 0.004$), which indicates that experienced rats have a stronger attention focus compared to naïve rats, even when they require the same number of trials to learn the task rule. Average angles at cue onset were lower in experienced ($N = 51$) vs naïve ($N = 38$) rats: 10.4 (9–13.8) vs. 13.2 (11.1–16.3) ° ($p = 6 \times 10^{-4}$, two-sided Mann–Whitney test), indicating higher levels of attention in experienced rats.

Second, it has been proposed that transfer effects during human rule learning occur because subjects tend to reuse pre-existing strategies, rather than relearn all possible state-action mappings[3]. This is another type of strategy-based dimensionality reduction. This scenario is compatible with the ACL model but adds another computational layer: it offers an explanation of how strategies and attention focus are selected in the first place. Such hierarchical models assume that strategies are selected from a pre-existing repertoire of hypothetical rules based on their value or created anew[3,27].

A hierarchical framework predicts that previously reinforced strategies are chosen earlier in a novel context because they have a higher value. We compared strategy selection between two cohorts of rats receiving random reward feedback that were either naïve or that had previously learned three different experimenter-defined rules (go-dark→ place→ alternate). Attention-at-choice was not different between cohorts, but experience indeed had an effect on how often specific strategies were selected (Supplementary Fig. 7a). The strategies go-dark and alternate were chosen earlier in the cohort with prior learning experience (Fig. 3e). Further, if strategies are re-selected as a whole based on their value, both negative and positive transfer effects can be expected[3]. Indeed, learning the rule go-silent after a sequence of three experimenter-defined rules (go-dark→ place→ alternate) was slower if rats were perseverating on the previously reinforced strategy alternate (Fig. 3f). In contrast, the number of times rats used the strategy alternate in previous task rules (which indicates a higher strategy value) predicted how fast they inferred the task rule alternate (Fig. 3g). However, many rats did not use the strategy alternate before it was reinforced (Fig. 3g). This indicates that some strategies are not part of their repertoire (see also Supplementary Fig. 7a) but are created anew. We provide evidence for strategy formation in the Supplementary Information (Supplementary Note 1, Supplementary Fig. 7b–d).

## Abstract representations of strategies and attention in PFC

Research on hierarchical rule learning in humans also suggests that strategies as hypothetical rules, are represented in prefrontal brain activity[3]. We tested whether such abstract prefrontal representations of strategies exist, whether they are related to selective attention, and how they compare to the decoding of other task variables. Using multiple single-unit recordings, we decoded behavioral strategies from the activity of prelimbic neurons in rats performing rule switches in either the novel multidimensional rule-learning paradigm or a conventional set-shifting task (62 sessions from nine rats, 34 (21–38) neurons per session; Methods, Fig. 4a). Maximum decoding accuracy from population activity for each pair of strategies ($N = 105$) within an experimental session was $80.9 \pm 1.0$% and peaked at 0.66 s (−1.05 to +1.2) after lever onset. Interestingly, decoding accuracy was already above chance 3 s prior to cue onset ($72.5 \pm 1.2$%, $p = 2.7 \times 10^{-36}$, one-sample t-test; Fig. 4b). We also conducted a population analysis for the entire 20 s ITI (moving window of 3000 ms width with step size of 300 ms) and found that the minimum value for decoding accuracy was $64.3 \pm 1.2$ % ($N = 105$ strategy pairs) which occurred 10.2 (6.68–14.1) s

prior to cue onset. This was significantly lower than for the 3 s prior to cue onset ($1.1 \times 10^{-25}$, two-sided paired t-test) but higher than the 50% chance level ($8.9 \times 10^{-21}$, one-sample t-test). Electrophysiological findings thus corroborate behavioral results, which indicate that action selection is a top-down process and not tied to specific cues.

Further analyses showed that population-level decoding depends on small contributions from many strategy-selective units. Prefrontal units typically have a low signal-to-noise ratio (SNR) but are often responsive to multiple task features, i.e., they exhibit mixed selectivity[34,35]. We tested whether single-unit firing significantly discriminated between either different strategies, side of lever press, reward feedback or location of visual and auditory cues. 67.8% of all units were responsive to at least one task feature (Bonferroni-corrected two-sided, unpaired t-tests with $p < 0.05$; Supplementary Methods). The firing rate of 37.1% of units discriminated between at least one pair of strategies in at least one task phase (46.4% also prior to cue onset; Fig. 4c), which is in line with literature on prefrontal rule representations in rats[13] and primates[36,37]. Unit firing was also significantly modulated by the side of lever press (40.9%), reward feedback (28.9%), or sensory cue location (visual cue: 11.9%, auditory cue: 7.6%). Moreover, many units were responsive to multiple task features (36.8% of units responded to ≥2/5, 15.1% to ≥3/5 of the tested task features; see Fig. 4c, Supplementary Fig. 8a–c).

We further computed the selectivity index d′ (Supplementary Methods) for strategies during three trial phases for 3348 strategy pairs from 1884 units. Consistent with previous literature on rule learning[13], d′ values were generally low SNR for units (pre cue: 0.2 (0.1–0.34), post cue 0.21 (0.1–0.36), post lever 0.22 (0.1–0.39); Supplementary Fig. 8d). To examine how single-unit selectivity is related to population decoding, we performed two analyses. First, we compared population-level decoding based on all units in a session with accuracy after sequentially removing one unit at a time. On average, the resulting difference in decoding accuracy was small in all trial phases (pre cue: −0.27 (−1.4 to +0.87) %, post cue: −0.27 (−1.53 to +0.93) %, post lever: −0.2 (−1.27 to +0.8) %; Supplementary Fig. 8e). Linear regression confirmed that accuracy differences were predicted by d′ in all three trial phases but other regressors in the model were not consistent predictors (Supplementary Methods, Supplementary Fig. 8f). Second, we ranked all units in a session according to d′ in each trial phase and repeated population decoding after removing the top 5, 10, 20, 30 or 40% units from the decoding analysis. Only after removing the top 40% of units, decoding accuracy was not higher than a 50% chance level (one-sample t-test with $p > 0.05$), and this effect was restricted to the respective trial phase (Supplementary Fig. 8g).

Given the strong behavioral evidence for the close connection between selective attention and strategy-based learning, we tested whether attention scores for a pair of strategies correlated with population decoding accuracy. There was a significant correlation between the median attention-at-choice score and decoding accuracy in the 3 s following cue onset ($p = 0.006$) after correcting for the number of neurons recorded in this session ($p = 1.7 \times 10^{-5}$, $R^2_{adjusted} = 18.2$ %; Fig. 4d). The correlation between attention-at-reward and decoding accuracy in the corresponding trial phase was not significant ($p = 0.09$) after correcting for the number of neurons recorded ($p = 2 \times 10^{-4}$, $R^2_{adjusted} = 11.8$ %).

To quantify how prefrontal representations of strategies compare to representations of other task variables, we conducted two analyses. First, we evaluated how the normalized decoding score (Supplementary Methods) of strategies and the current attention focus (Fig. 2a) compares to that of cues and specific actions in consecutive trial phases. A two-way repeated-measures ANOVA had a significant interaction effect ($N = 60$ sessions, $F_{(6,54)} = 32.65$, $p < 10^{-3}$, multivariate results) and we found significant simple main effects for all task stages (Pre cue: $F_{(3,57)} = 85.5$, Post cue: $F_{(3,57)} = 90.6$, Post lever: $F_{(3,57)} = 99.6$, all $p < 10^{-3}$). Bonferroni-corrected two-sided pairwise

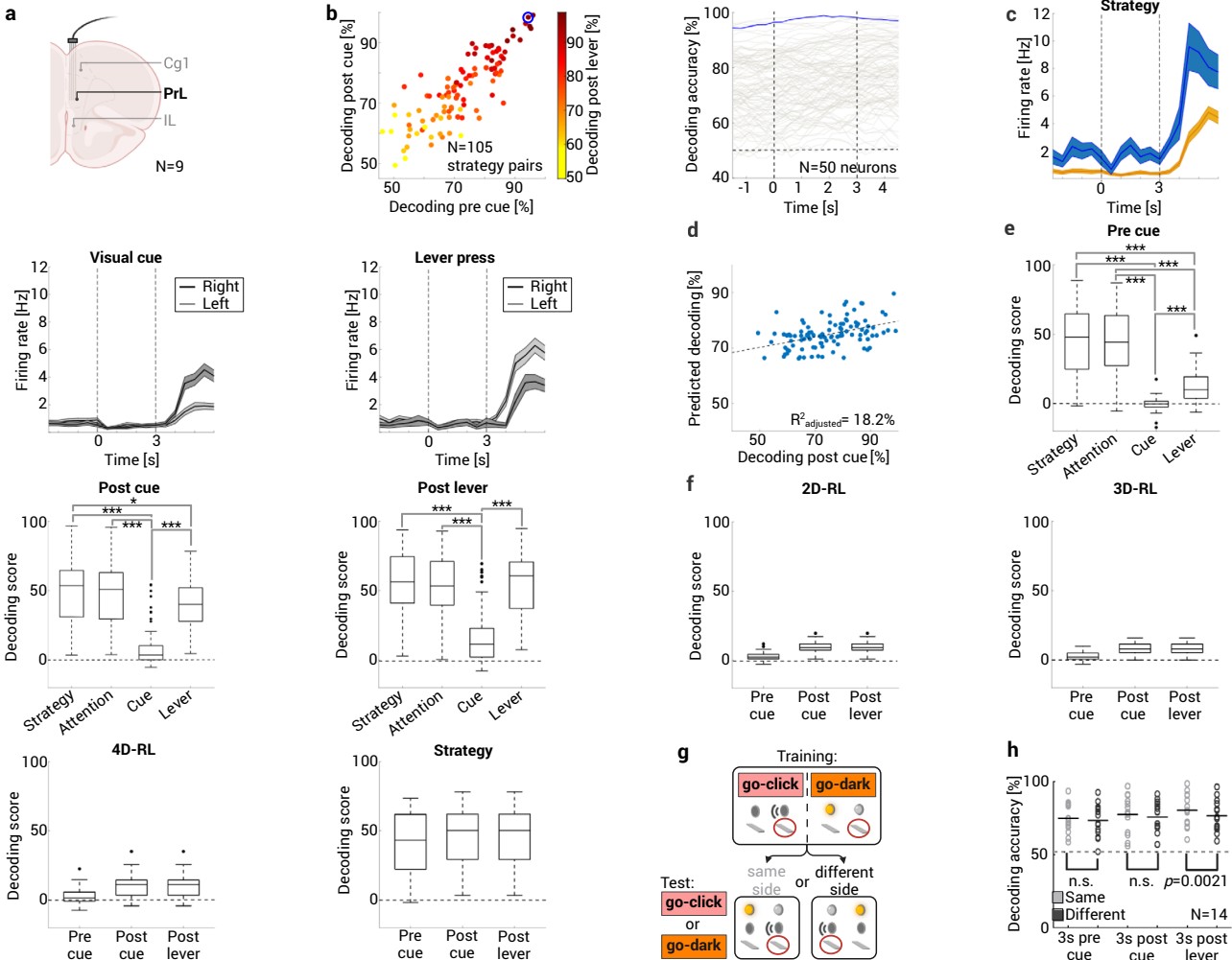

**Fig. 4 | Abstract representations of strategies and attention in PFC. a** Population decoding based on prelimbic (PrL) multiple single-unit activity. Cg1/cingulate cortex, IL/infralimbic cortex. **b** *Left:* decoding accuracy for three consecutive trial phases of each strategy pair. Above-chance decoding before cue onset indicates top-down action selection. *Right:* decoding across time, one strategy pair is highlighted (marked in left panel; dotted horizontal line: 50% chance level, vertical lines: cue and lever onset). **c** PSTH of a unit (from highlighted session in **b**) with 1.3 Hz mean firing rate that significantly discriminated (Bonferroni-corrected, two-sided unpaired *t*-tests) between strategies (*go-dark*/orange, *go-left*/blue; $p_{pre\ cue} = 6.4 \times 10^{-5}$, $p_{post\ cue} = 3.5 \times 10^{-7}$, $p_{post\ lever} = 4.1 \times 10^{-7}$), visual cue location ($p_{post\ lever} = 3.3 \times 10^{-5}$), and side of lever press ($p_{post\ lever} = 3.5 \times 10^{-3}$). Data presented as mean values ± SEM. **d** Scatter plot of regression model: median attention-at-choice predicted decoding post cue in a session (corrected for number of neurons/session). **e** Two-way repeated-measures ANOVA to evaluate decoding of strategies, attention focus, cues, and specific actions across trial phases. Decoding of strategies and attention focus is comparable and outperforms decoding of cues and actions with most prominent

effects before cue onset ($N = 60$ sessions, Bonferroni-corrected pairwise comparisons, ***$p < 10^{-3}$; chance level: normalized decoding score of 0). **f** Two-way repeated-measures ANOVA to compare decoding of strategies vs. state-action-pairs as defined in the 2D-RL, 3D-RL and 4D-RL models (Fig. 2d). Strategy decoding outperforms all other models ($N = 39$ sessions, Bonferroni-corrected pairwise comparisons, all $p < 10^{-3}$; chance level: normalized decoding score of 0). **g** Prefrontal strategy representations did not depend on motor response. A classifier was trained on trials with lever presses on one side, and decoding accuracy was compared when tested on trials with lever presses on either the same or opposite side. **h** Accuracy was not different between conditions in the 3 s before and after cue onset, but significantly lower in the 3 s following lever onset (dot plots with mean as horizontal line, two-sided paired *t*-test). Decoding accuracy was above chance in all conditions (all $p < 10^{-5}$, two-sided unpaired *t*-test; dotted horizontal line: 50% chance level). Box plots showing median, 25%–75% percentile, whiskers: 1.5 IQR, and outliers. Created in BioRender. Böhme, N. (2025) https://BioRender.com/v10a089. Source data are provided as a Source Data file.

comparisons showed that decoding of strategies and current attention focus is comparable and outperforms decoding of cues and simple actions, with most prominent effects appearing prior to cue onset. Decoding of cues was weakest in all follow-up comparisons (all $p < 10^{-3}$) and at the time of lever press, decoding accuracy was not different between strategies, attention focus and simple motor actions (Fig. 4e). Second, we conducted another two-way repeated-measures ANOVA (interaction effect not significant: $F_{(6,33)} = 2.3$, $p = 0.056$, multivariate results) to test how decoding of strategies across three trial phases compares to decoding of s-a pairs as defined in the 2D-RL, 3D-RL and 4D-RL models shown in Fig. 2d ($N = 39$ sessions). There was a main effect of the factors model ($F_{(3,36)} = 49.4$, $p < 0.001$, multivariate

results) and trial phase ($F_{(2,37)} = 75.9$, $p < 10^{-3}$). Bonferroni-corrected two-sided pairwise comparisons showed that conceptualizing actions as strategies was better than all other models (all $p < 10^{-3}$) but the three classical RL models were not statistically different from each other. Moreover, action decoding was highest at lever onset and lowest prior to cue onset (all $p < 10^{-3}$; Fig. 4f).

Moreover, we evaluated whether there is neural evidence that strategies are abstract actions as opposed to simple motor responses[38]. To test this, we trained a classifier to discriminate between strategy pairs, but only using trials where all lever presses were on one side (i.e., all trials with either left or right lever presses for both strategies). We then compared strategy decoding in two testing conditions: either

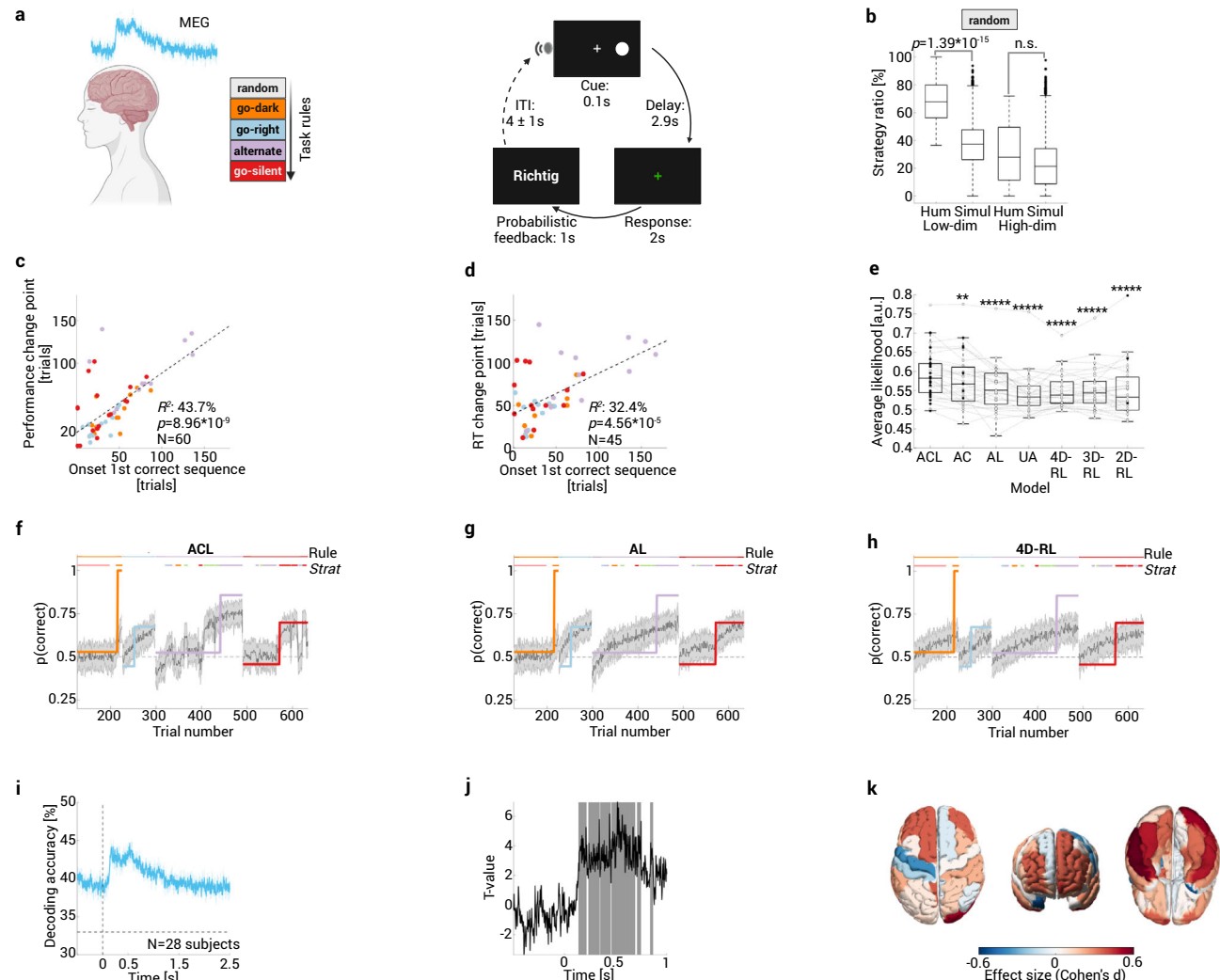

**Fig. 5 | Humans also infer task rules using low-dimensional strategies. a** Healthy adults solved an adapted version of the multidimensional rule-learning task during MEG recordings. *Left:* order of task rules. *Right:* trial structure. ITI/inter-trial interval. **b** Higher strategy ratio for low-dimensional strategies in humans performing the random rule vs. a random agent (two-sided Mann–Whitney test). In contrast, the ratio for high-dimensional strategies was not higher than chance (two-sided Mann–Whitney test). The ratio was higher for low- vs. high-dimensional strategies in humans ($1.58 \times 10^{-6}$, two-sided Wilcoxon signed rank test). **c** Significant Pearson correlation between the onset of the first correct strategy sequence and the performance change point mirrors rat findings. **d** Significant Pearson correlation between the onset of the first correct strategy sequence and a decrease in reaction time. **e** Replication of RL model findings in humans. Dashed lines connect values (circles) of the same subject; filled circles represent the best (i.e., highest cross-validated likelihood) model of each subject. Model comparisons between ACL and all other models are Benjamini–Hochberg corrected two-sided Wilcoxon matched pairs test with **$p < 0.01$, ****$p < 10^{-4}$, *****$p < 10^{-5}$. **f–h** Representative example of

simulations using model parameters estimated from experimental data of one human. Color-coded sigmoidal learning curves show transitions instead of gradual learning. Sudden performance changes could only be reproduced if choice was modulated by strategy-specific attention (ACL in **f**), but not AL (**g**) or 4D-RL models (**h**). **i** Trial-based decoding of current attention focus across time (similar to rats; Fig. 4e) is shown with respect to cue onset (vertical dashed line), 33% corresponds to decoding at chance level (horizontal dashed line). **j** Decoding increased following cue onset (cluster-based permutation test[69], comparison against pre-stimulus baseline). Shaded areas correspond to time points for which the H0 hypothesis of no difference between decoding was rejected. **k** Source-level topography of attention decoding during task (0–1000 ms) vs. baseline (−500 to 0 ms). Note increased attention decoding following cue onset in task-related sensory areas and in PFC (two-sided, paired *t*-test; *t*-values converted to Cohen's *d*). Box plots showing median, 25%–75% percentile, whiskers: 1.5 IQR, and outliers. Created in BioRender. Böhme, N. (2025) https://BioRender.com/w17o467. Source data are provided as a Source Data file.

using trials where all lever presses were on the same side used for training, or on the opposite side (different condition; Supplementary Methods, Fig. 4g). If the PFC encodes simple motor actions instead of abstract strategies, we would expect to find a significant decrease in decoding accuracy for the different condition. Instead, we found that the side of the lever is not relevant in the 3 s prior to and following cue onset. The side of the lever was only relevant for strategy decoding when the action is performed, indicating that this representation is not driving low-level action selection (Fig. 4h). Further, we found neural representations of the task feature that is relevant for the current task rule before learning was detectable in behavioral choice data

(Supplementary Note 2, Supplementary Fig. 9). In sum, we detected an abstract neural representation of specific strategies and the associated current attention focus throughout trial phases, which is in line with their proposed role in top-down action selection.

**Humans also infer task rules using low-dimensional strategies**

Although our rodent task is inspired by concepts from human research, it is unknown whether humans show similar behavioral and neural processes in this specific experimental context. We therefore analyzed data from 31 healthy adults who solved an adapted version of the multidimensional rule-learning task during MEG recordings (Fig. 5a).

Due to limits on MEG session duration, the number of trials was necessarily lower in the human task version. Nevertheless, subjects tested most of the behavioral strategies (6 (5–6) out of 8 strategies) and were able to learn 58.1% of the rules. Performance in our task could be related to established markers of executive function as measured using the CANTAB neuropsychological test battery that all subjects completed (Supplementary Table 9).

Similar to what we observed in rats, the median sequence length was 15 (12–18) trials. As in rodents, the detected strategies are unlikely to be false positives: the strategy ratio was significantly higher in humans performing the random rule ($N = 31$) vs. a simulated randomly behaving agent (10,000 sessions; Methods, Fig. 5b). Similar to what we found in rats (Fig. 1c), there was also no evidence that humans used alternative, higher-dimensional strategies during rule learning. There was no statistical difference between the strategy ratio for high-dimensional strategies in humans vs. a random agent ($p = 0.13$, two-sided Mann–Whitney test; Fig. 5b). Moreover, there was a significant difference for the strategy ratio in humans for low- vs. high-dimensional strategies ($1.58 \times 10^{-6}$, two-sided Wilcoxon signed rank test).

We also provide several lines of evidence that strategies are related to rule learning. Across rules, the onset of the first correct strategy sequence correlated with both increases in performance (Fig. 5c) and decreases in reaction time (Fig. 5d). Interestingly, the onset of the first correct strategy sequence occurred before the decrease in reaction time (trial 29 (15–63) vs. 51 (40.5–86.5), $p = 1.2 \times 10^{-3}$, two-sided Wilcoxon matched pairs test). Faster responses, indicating higher confidence in their choices, thus occurred only after subjects tested the correct strategy for the first time. This supports the idea that humans indeed test strategies to identify the correct task rule.

Moreover, we investigated evidence for strategy-based learning further by fitting our attention-modulated and standard RL models to human behavior. As in the rat cohort, we found that the binary ACL model was the best at predicting held-out behavioral data in 19 out of 31 subjects, followed closely by the AC model in 9 out of 31 subjects (Fig. 5e). Again, model simulations constrained by human experimental data showed that experimentally observed sudden changes in performance could only be reproduced if choice was modulated by the strategy-specific attention focus (Fig. 5f–h). As in rats, ACL model variants in which strategy-specific attention effects for either auditory, visual, or place-outcome strategies were removed, resulted in significantly worse prediction of held-out behavioral responses (Supplementary Fig. 6e).

In our main analysis at the neural level, we decoded the current attention focus instead of directly decoding strategies. This was necessary for the (random-effects) group-level analysis because subjects did not all use the same strategies but sampled the same task features (e.g., go-click/go-silent share the same task feature). As for animals (Fig. 4e), decoding of the current attention focus was already above chance *prior* to cue onset (i.e., the lower confidence interval of the curve exceeded chance level at all time points; Methods, Fig. 5i, Supplementary Fig. 10a). This indicates that attention allocation in humans is also driven by top-down processes related to strategy learning rather than being tied to specific cues. Decoding accuracy significantly increased in the second following cue onset as compared to the pre-stimulus baseline (Fig. 5j). We used multivariate pattern searchlight analysis to characterize the source-level topography (Methods) of attention decoding during task (0–1000 ms) vs. baseline (−500 to 0 ms). Note increased attention decoding following cue onset in task-related sensory areas (auditory, visual) and in PFC (Cohen's d; Fig. 5k). We conducted an additional source-level fixed effects analysis to evaluate whether there is evidence for neural representation of strategies in humans. Indeed, trial-based decoding across time was above chance (although only following stimulus presentation; Supplementary Fig. 10c) and the peak was located in the right cingulate

(Supplementary Fig. 10d). In sum, behavioral and neural findings in humans replicate those in rodents, suggesting that both species address novel task challenges by probing different low-dimensional behavioral strategies, in contrast to solving the higher-dimensional problem of learning all possible cue-action pairs.

## Discussion

Identifying the computational processes of executive functions and their neural implementation[39,40] is crucial to gain a mechanistic understanding of cognitive flexibility and how it is impaired in psychiatric conditions[1,41]. However, a key challenge on that path is the explanatory gap between macroscopic human brain imaging and the microcircuit level in animals[39,42]. A cross-species approach has been proposed to overcome this barrier that includes aligned task paradigms in both species with detailed computational modeling of cognitive processes and the use of complementary neural activity measures that can be compared in a common representational space[39,42–44]. However, careful behavioral work is central to interpreting neural data[45,46]: neural representations are only informative if we can make sure that cross-behavioral assays engage similar cognitive processes. This is important because most learning problems can, in principle, be handled by many different computational mechanisms[40] and humans have remarkable abilities to learn in complex environments[16] that may be qualitatively different from how animals solve the same task[47]. Therefore, a key aspect of our study is that it goes beyond performance measures used in traditional rule learning paradigms[1] and evaluates whether there is a conserved set of computational mechanisms at work when both species are challenged in a matched paradigm. Several computational models can explain how humans infer task rules in complex environments[3,5,16,27]. In rodents, however, these learning mechanisms are largely unknown in spite of a wealth of information about task-related neural firing patterns, necessary brain regions, and involved neurotransmitters in the context of rule-switching paradigms[33,41]. We therefore designed rat experiments to test behavioral and neural predictions made by different learning models and then evaluated whether these computational processes are also at work in a matched paradigm in humans.

The aim of our study was twofold. First, we tested our core hypothesis that both rats and humans sequentially follow different hypothetical rules—which we call *behavioral strategies*—to identify the current task rule. The existence of behavioral strategies in rats and humans is well-documented, and it has been speculated that it could serve as an exploratory behavior that samples environmental cues for relevance, but the precise role in higher-order cognition is less clear[28–30,48]. Despite huge differences in learning speed (see Fig. 1e vs. Fig. 5c), results from behavioral and computational modeling indicated that low-dimensional strategies predict learning in both species and are able to explain why sudden transitions in performance occur (Figs. 1d–f, 2c–g and 5c–h). Our results also indicate that low-dimensional strategies provide a better explanation for behavioral responses than random choice, high-dimensional strategies, and learning the full state-action-space or variants thereof (Figs. 1c, 2d and 5b, e).

Second, we used RL models and tested their predictions on behavioral and neural data to understand which computational processes are species-conserved[3,5,11,23]. Our working hypothesis (Fig. 6) based on our data is that there are two levels at which the task is performed: (1) a lower level at which state-action values are learned within the attended-to sub-state space, and (2) a higher level at which strategies are selected from a pre-existing repertoire or created anew.

Attention can reduce the perceived complexity of a multi-dimensional environment and allows for efficient sampling of a simplified state space[11]. In line with previous literature[49], we found head movements related to cue onset and reward feedback, and these orientation reactions were modulated by strategy-specific attention

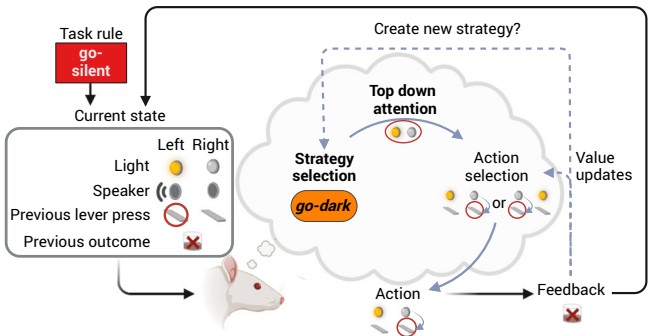

**Fig. 6 | Schematic of computational processes underlying rule inference.** Value-based strategy selection and top-down attention affect decision making and learning in a trial. Our working hypothesis, based on our data, is that there are two levels at which the task is performed: (1) A lower level at which state-action values are learned within the attended-to sub-state space. This sub-space corresponds to the *current attention focus*, where strategies map to distinct low-dimensional state spaces via their attention focus. (2) A higher level at which strategies are selected from a pre-existing repertoire of *hypothetical rules* based on their value or created anew. Strategies are abstract because they summarize different combinations of environmental cues, actions, and potential outcomes within a common concept (in contrast to simple motor actions or cue-driven action selection), and this facilitates the transfer of learned skills to new tasks. Both computational layers contribute to strong dimensionality reduction during strategy-based learning. We model the lower level using attention-modulated RL models, and predict the existence of the higher level based on observed transfer effects (value-based strategy selection) and neural decoding analyses (abstract prefrontal representations). Despite differences in learning speed across species (and additional human-specific computational processes[16]), our results indicate that these computational mechanisms are shared with humans. Created in BioRender. Böhme, N. (2025) https://BioRender.com/wex0oga.

(Fig. 2b, Supplementary Figs. 2–4). Stronger attention-at-choice predicted learning (Fig. 3d), and task responses in both species were best explained by the ACL model (Figs. 2d, 3b, c and 5e), similar to what has recently been described in humans[11]. Furthermore, there was no difference between attention-modulated RL models based on either attention scores measured by head direction or binary attention scores in rodents. The latter corresponds to the scenario that the rat is attending fully to the current task cue, which indicates that attention is a very strong filter and is thus additional evidence for low-dimensional strategies. Note that both ACL models are equivalent to sensory filtering models where selective attention acts on sensory inputs and thereby influences both choices and learning[50,51]. Interestingly, similar attentional mechanisms also underlie the flexibility and generalization capabilities in powerful state-of-the-art AI systems based on Transformers[52–54].

The higher level addresses the question of how a strategy and the associated attention focus are selected, which is a challenge for attention-modulated RL models[5,11]. We tested behavioral and neural predictions of hierarchical RL, an influential model of human cognitive flexibility[3,27], to provide evidence for that second level. Note that serial hypothesis testing models are also relevant in this context, but these models assume that hypothesis testing occurs across a fixed set of behavioral strategies (i.e., strategies are not formed de novo)[55,56]. At the behavioral level, rats preferably selected previously reinforced strategies in a novel context (Fig. 3e, Supplementary Fig. 7a) and added new strategies to their repertoire based on task demands (Supplementary Note 1, Supplementary Fig. 7b–e). Note that the exploration-exploitation balance is also relevant for action selection at the higher level, which explains why rats sometimes abandon a correct strategy (e.g., task rule alternate in Fig. 2e). We found that both strategy values and top-down attention contribute to transfer effects, another salient phenomenon observed during rule learning (Fig. 3). Future data-

guided computational modeling of high-level, strategy-based learning could complement our current low-level attention-modulated RL approach to provide novel, testable predictions on how subjects take advantage of regularities in the environment to generalize established behaviors to new situations[5,22].

Our behavioral results support the idea that similar cognitive processes are engaged in aligned task versions in rats and humans. We used neural decoding analyses to provide further evidence for shared computational processes during rule learning in both species. Our approach was inspired by an extensive body of work on neural representations of RL quantities such as action values or reward prediction errors in human fMRI and animal single-unit recordings[9,10]. Indeed, we showed that constructs related to the computational processes formalized by the RL models (specifically, strategies and the current attention focus; Fig. 6) can be reliably decoded from both rodent prefrontal single-unit activity and human prefrontal MEG (Figs. 4 and 5i–k, Supplementary Figs. 8 and 10). Of course, our analyses are restricted to this level of computational quantities represented in the prefrontal cortices of both species, as it is not possible to directly map macroscopic MEG to cellular resolution electrophysiology. Further evidence for the proposed role of low-dimensional strategies in top-down action selection was provided by above-chance decoding before stimulus onset in both rats and humans.

Our human findings are consistent with the proposed role of lateral PFC representations during task rule inference[5,48]. Nevertheless, a translational link remains challenging because large parts of the human PFC probably have no clear corresponding homologs in rodents[57]. However, individual prefrontal neurons in both rodents and primates have consistently been found to encode abstract task rules in the steady state (i.e., *after* task rules have been reliably identified), indicating that some neural computations are conserved[13,36,37].

For our conclusions, it was important to exclude that prelimbic neurons encode body position rather than strategies per se, which is a known confounder in prefrontal recordings from freely-moving rodents[58]. This was supported by the observations that PFC population activity represents abstract strategies rather than simple motor actions (Fig. 4g, h) and that strategies could be decoded throughout all trial phases even though rat orientation and location in space was highly variable both within and across trials (Fig. 4b). Furthermore, attention-at-choice indeed affected prefrontal strategy decoding (Fig. 4d). While this provides evidence for the link between strategy selection and current attention focus, it also shows that it is not possible to disentangle movement from cognition completely because our behavioral read-out for attention is the orientation reaction following cue onset.

Our results highlight the potential of an aligned, yet complementary approach in both species. For example, higher neural SNR in rats allowed us to show that prefrontal computations may support generalization across contexts because they reflect only the shared structure in the environment with unnecessary details discarded (Fig. 4g, h, Supplementary Fig. 9)[59]. In contrast, human whole-brain imaging has lower SNR and spatial resolution, but it was possible to identify the entire cortical network (not restricted to PFC) related to the current attentional focus (Fig. 5k). Applying such a cross-species approach is not only relevant for basic neuroscience but may also increase predictive validity in animal models of neuropsychiatric diseases[42]. In turn, these insights could inform studies in clinical populations that combine empirical attention measures (e.g., using eye tracking), imaging and computational approaches to better understand how altered neural representations lead to impaired cognitive flexibility[40].

## Methods
### Animals
Two hundred sixteen male Sprague Dawley rats (Charles River, Germany) were 6 weeks old when they arrived at our animal housing

facility. They were group-housed in standard Macrolon cages ($55 \times 33 \times 20$ cm) with free access to drinking water throughout the study. A subset of rats ($N = 15$) was implanted with silicon probes, and these rats were single-housed after surgery in the same cage type with a custom-made lid to prevent implant damage. Food was available *ad libitum* for the first 2 weeks. Thereafter, food was restricted to ~20 g per rat per day which maintained them on a stable bodyweight. Lights were turned on from 7:30 am to 7:30 pm and experiments were performed during the light phase. At the start of experiments, rats were at least 8 weeks old. The head and neck of rats (except for implanted rats) were marked with a black hair dye (Koleston Perfect 2/0, Wella, Paris, France) to facilitate the detection of body parts in video analyses. All experiments in this study were performed in accordance with national and international ethical guidelines, conducted in compliance with the German Animal Welfare Act and approved by the local authorities (Regierungspräsidium Karlsruhe, Germany, approval numbers 35-9185.81/G-4/16, 35-9185.81/G-133/20). Efforts were made to reduce the number of animals used, and all behavioral protocols were refined to minimize adverse effects on animal well-being. Throughout the study period, no adverse health events occurred that demanded special veterinary care or removal of an animal from any experiment. Seventeen rats were excluded from data analysis either due to incomplete data (hardware/software problems or human error, $N = 12$) or bad signal quality in the recovery period after surgery in the case of implanted rats ($N = 5$). Supplementary Table 7 provides an overview of how the remaining 199 rats were distributed across experimental groups.

## Human participants

32 healthy adults without a history of mental disorder (22 females/10 males, median age 24.5, range 19–55 years) were recruited from the local community. Participants had normal or corrected-to-normal vision. All subjects gave written informed consent and received a financial compensation of 50€ for participation. The participants were invited to two appointments. The first one consisted of an MEG session during which participants performed the multidimensional rule-learning experiment. The second consisted of an anatomical magnetic resonance imaging (MRI) scan and a 90-min neuropsychological test battery (CANTAB, Cambridge Cognition, Cambridge, UK). One subject discontinued participation in the study prematurely; three subjects could not be included in the MEG analysis (two due to strong artifacts, one subject for behavioral reasons that are detailed in the section on MEG analysis). The study was approved by the local ethics committee (Ethik-Kommision II, University of Heidelberg, Medical Faculty Mannheim, Germany, approval number 2020-568N).

## Rat operant procedures

Initial operant training took place in automated operant training chambers ($20.5 \times 24.1$ cm floor area, 29.2 cm high; MED Associates, St. Albans, VT, USA) that were equipped with two retractable levers, located left and right from a central food tray. Cue lights were located above each lever, and a house light was placed in the upper left corner opposite the food tray. All chambers were light- and sound-attenuated, and a ventilator provided constant background noise. All procedures were controlled by a computer running custom-made MedStat notation code (MedPC IV, MED Associates, St. Albans, VT, USA). Initially, rats were trained to respond equally to the presentation of both levers individually[33]. Rats were exposed to the experimental stimuli before the actual experiment started (i.e., sensory stimuli were not novel). The rule-learning tasks were performed either in the same boxes (operant two-rule set-shifting task with deterministic reward feedback) or in large ($30 \times 48 \times 41$ cm, custom-made) operant boxes (all implanted rats and rats performing the multidimensional rule-learning task). For the multidimensional rule-learning task, operant chambers were additionally equipped with two loudspeakers located above each lever. We used either sweetened condensed milk (small boxes: 80 µl, Milchmaedchen, Nestlé, Germany) or food pellets (big boxes: 45 mg food pellets, BioServ F0021, Flemington, NJ, USA) as rewards.

## Rat multidimensional rule-learning task

A pseudorandomized list was used for presenting one visual and one auditory cue (click sound) at the beginning of each trial (Supplementary Table 8). Three seconds after trial onset, two levers were presented. Rats had to respond within 10 s after presentation, otherwise the trial was considered an omission trial. At the end of the trial, the levers were retracted, and the next trial was started after a fixed ITI of 20 s. Each session consisted of 300 trials (i.e., independent of performance) and no rule switches occurred within a session. Nine different experimenter-defined rules (go-light, go-dark, go-click, go-silent, alternate, go-right, go-left, win-stay-lose-shift, random) were used. The random rule refers to a session with random reward feedback. Rats learned either a single rule or one of two versions of a fixed sequence of four rules. The go-left rule was excluded for rats learning only one rule. The sequence go-dark→ place→ alternate→ random ($N = 11$) was used to test the effect of prior learning on strategy selection in the last rule. Five random rule sessions were performed and compared to the first five sessions of a separate cohort of naïve rats ($N = 19$) also receiving random reward feedback. The sequence go-dark→ place→ alternate→ go-silent ($N = 22$) was used to investigate positive transfer effects (e.g., accelerated learning of the go-silent rule as the fourth versus the first rule). In both four-rule task versions, the *place* strategy (i.e., always pressing the right or the left lever) that rats spontaneously used less during learning of the go-dark rule was selected as the second experimenter-defined rule for each rat (go-right: $N = 22$, go-left: $N = 11$). In multi-rule experiments, uncued rule switches occurred after the percentage of trials explained by the correct strategy (i.e., the one corresponding to the task rule) exceeded 50% of all trials in the session or if a rat reached the criterion of 18 correct trials out of 20 in two consecutive sessions. Reward feedback was deterministic except for the random rule (50% reward independent of choice), and win-stay-lose-shift (80% reward for correct responses, 20% reward for incorrect responses; otherwise, it would not be possible to tell whether a rat followed *win-stay-lose-shift* or a *place* strategy).

## Rat surgery

Fifteen rats underwent surgery for microelectrode implantation after they had learned to respond equally to the presentation of both levers individually. 64-channel silicon probes (chronic P1-probe with four shanks and 16 channels/shank; Cambridge NeuroTech, Cambridge, UK) attached to a drive (nano-Drive; Cambridge NeuroTech, Cambridge, UK) were implanted into the right prelimbic cortex (center of probe placed at: AP +3.0, ML +0.6, DV −3 mm from brain surface) of rats anesthetized with isoflurane (2.0–2.5%). A bone screw above the cerebellum served as ground. Electrodes were only moved if signal quality declined. For each rat, placement of electrodes within the prelimbic cortex was confirmed using histological methods. More specifically, animals were deeply anesthetized and transcardially perfused with a 4% buffered formalin solution. The entire head with electrodes in place was kept in formalin solution for 3 weeks before the brains were collected and then sectioned using a vibratome. This procedure ensured that electrode tracks were visible without further staining.

## Rat electrophysiological recordings

Rats were allowed to recover for at least 7 days after surgery before they were accustomed to the recording set-up and performed additional training sessions. The actual rule-learning paradigms started at least 14 days after electrode implantation. Multiple single-units were simultaneously recorded using a 64-channel RHD2164 amplifier connected to a RHD2000 USB interface board (Intan Technologies LLC,

CA, USA). Channels were digitized with 16-bit resolution, sampled at 30 kHz, and bandpass filtered between 0.1 and 7500 Hz. Time stamps for cue lights, lever presentation, and lever presses were transmitted from the Med Associates behavioral control system to the Intan recording system to align behavioral markers with neural activity.

## Human rule switching

We developed a version of the multidimensional rule-learning task during MEG recordings using Presentation (Version 20.1, Neurobehavioral Systems, Berkeley, CA, USA). We aimed for a task that is both close to the rat multidimensional rule-learning task (i.e., same stimulus material, sequence of multiple, uncued rule switches) and can be completed in a single MEG session. This came at the cost of having fewer trials as compared to the rat data. At the same time, we aimed to strike a balance between the level of difficulty (to avoid a ceiling effect) and being able to perform several rule switches in one session. Based on a behavioral pilot outside the scanner, we used a fixed number of trials/rule (as opposed to performance-dependent rule switches in rats), and reward feedback was probabilistic (80% for correct, 20% for incorrect responses) to make the task more challenging (deterministic reward feedback in rats). Moreover, the human task version had an additional task rule with randomized reward feedback at the beginning to examine whether hypothesis testing reflects a general cognitive strategy, even when there is no behavioral advantage similar to what we observed in rats (Figs. 1c and 5b). The experiment comprised 650 trials and consisted of the following five different experimenter-defined rules: random (126 trials)→ go-dark (100 trials)→ go-right (75 trials)→ alternate (199 trials)→ go-silent (150 trials). The trial structure was as follows (Fig. 5a): each trial started with the pseudorandomized presentation of a white circle and a sound on either the right or left side. The stimuli were presented for 0.1 s, followed by a 2.9 s delay interval with a white central crosshair. Participant responses (either right or left button press) were logged while the crosshair turned green for 2 s. Feedback then appeared on the screen for 1 s - either the word *richtig* (correct) as positive feedback or a white central crosshair as negative feedback. The white central crosshair was then displayed for 4 ± 1 s before the next trial commenced. Before the experiment, subjects were informed about the trial structure and told that their goal was to get as many correct answers as possible. Afterwards, participants were asked whether they identified the experimenter-defined rules and how they solved the task.

## Human MEG acquisition

MEG was recorded at a sampling rate of 1000 Hz using a 306-sensor TRIUX MEGIN system (MEGIN, Finland) in a magnetically shielded room (hardware filtering: 0.1–330 Hz). Signals were acquired by 102 magnetometers and 204 orthogonal planar gradiometers at 102 different scalp positions. A signal space separation algorithm implemented in the Maxfilter program provided by the manufacturer was used to remove external noise (e.g., 16.6 Hz train power supply and 50 Hz power line noise) and to align the data to a common standard head position across acquisition sessions based on the measured head position at the beginning of each session. Each participant's head shape and fiducials (nasion and pre-auricular points) were digitized using a Polhemus Fastrak digitizer (Polhemus, Vermont, USA). Continuous tracking of head position relative to the MEG sensors was achieved utilizing five head position indicator coils. Oculomotor events such as blinks and saccades were recorded using conventional vertical and horizontal electrooculography/EOG. EOG electrode impedance was kept below 10 kΩ.

## Strategy detection algorithm

For each strategy $i$, we identify trial blocks where following that strategy is more likely than chance or than following each of the seven other strategies. To do so, we first define eight binary time series $b_i$,

where $b_{ti} = 1$ indicates that behavior at trial $t$ is consistent with strategy $i$. For a candidate sequence length $n$, we compute count time series $c_{ti}(n) = \sum_{t-n+1}^{t} b_{ti}$, i.e., the number of trials within a window of size $n$, ending at trial $t$, for which behavior is consistent with strategy $i$, and estimate the probabilities within each of these $n$-blocks as $p_{ti} = c_{ti}/n$. We then evaluate the following binomial statistic (i.e., based on the binomial distribution) at every trial,

$$P_{tij}(n) = \Pr\left(c \geq c_{ti}|c_{tj}\right) = 1 - \sum_{c=0}^{c_{ti}-1} \binom{n}{c} p_{tj}^{c} \left(1 - p_{tj}\right)^{n-c} \qquad (1)$$

where $j$ indexes the alternative strategy $i$ is tested against. When testing against chance, we set $j = 0$, $p_{t0} = 0.5$ and $c_{t0} = n/2$. Strategy $i$ is significantly more likely than strategy $j$ for a block of size $n$ ending at trial $t$ when $P_{tij}(n) < 0.05$ (i.e., we ask the $H_0$ question: How likely is it to observe $c_{ti}$ or more trials consistent with strategy $i$ when the true strategy in place was $j$?). Starting from $n = 6$ (the minimum admissible sequence length at which this significance level can be achieved), sequence length is iteratively increased up to the number of trials in the session. If several overlapping blocks of a candidate strategy are significant, the block with the lowest binomial statistic when compared to the second most likely strategy is kept (usually the longest), and the others are discarded. Significant blocks where the animal does not follow the tested strategy in the first or last trial in the block are discarded, as well as blocks in which the strategy is not followed for more than two trials in a row. In cases where a significant, length $m$ block of one strategy $i$ is a subset of a significant, length $n > m$ block of another strategy $j$, both blocks are discarded to avoid false positives (the detection method is conservative). We excluded the strategy *win-stay-lose-shift* from strategy detection in experiments where rats learned a deterministic place rule.

We used the following approach to validate the strategy detection results. In order to quantify the rate of false negatives, we created sessions that consisted of 100 trials with random choices. Next, we inserted one synthetic strategy sequence per session at randomized positions and counted the percentage of strategy trials correctly detected by the strategy detection algorithm (i.e., the sensitivity of the detection algorithm). This process was repeated 10,000 times (1250 times for each of the eight strategies) for multiple sequence lengths (ranging from 6 to 50 trials). Moreover, we tested whether we can use choice data to exclude that strategy detection is not merely a false-positive result. For this, we directly compared the percentage of trials explained by strategies in the random rule against strategies detected in simulated data with randomized choices (10,000 sessions with the same length as in rats and humans). Finally, we evaluated whether strategy detection is specific for the proposed set of strategies or if similar results can be obtained for an alternative set of six strategies that could also be used to learn all rules (*win stay-lose shift* vs. *win shift-lose stay* approach with respect to place, visual, and auditory cues). We adapted the detection algorithm to compare the percentage of trials that can be assigned to high- vs. low-dimensional strategies in rats and humans performing the random rule and to compare it against high-dimensional strategies detected in simulated data with randomized choices.

## Rat video analysis

A night vision USB camera (ELP, Shenzhen, China) positioned above each operant chamber recorded animal behavior with 30 frames per second (Image Acquisition Toolbox, MATLAB, Natick, MA, USA). Video software (FFmpeg, https://www.ffmpeg.org/) converted the videos into gray scale, downsampled them to five frames per second, and separated them into single frames. Using open source software for machine learning-based image analysis (Ilastik-1.2.0, http://ilastik.org/)[60], frames were segmented into background, head and body of a rat. In each segmented frame, the center of the head and body was determined using a custom-written Python script (https://www.python.org/). Distances between the center

of the head and each of the two cue lights were determined. Additionally, the head angles with respect to both sides were computed by calculating the angles between the lines connecting head-to-body and the cue light to the body. A custom-written MATLAB script identified the frames corresponding to the cue light on- and offsets in the downsampled videos. Potential markers for attention during choice formation and reward feedback were identified as follows: we observed an orientation reaction after cue onset and used the minimum angle of view with the respect to the side of subsequent responses (in the second after cue onset) as a marker for attention at choice (Fig. 2, Supplementary Figs. 2–4, Supplementary Tables 1–3). Smaller angles $a_{AC}$ putatively correspond to a stronger attention focus. Similarly, rats often looked back and forth between the pressed lever and the food receptacle during reward feedback. We therefore assumed that the angle sum in the second after the lever press (volatility at reward $v_{AR}$) is a marker for attention-at-reward. A higher angle sum would correspond to higher attention levels. Based on careful screening of rat behavior in videos, we also evaluated the hypothesis that strategies can be identified based on movement patterns in the ITI (Supplementary Fig. 5). More specifically, we monitored the median position in the ITI (posITI: positive/negative values indicate that the animal is in the right/left compartment of the operant box) to test whether rats stay in the respective compartment while following a *place* strategy. For the strategy alternative, we tested if rats had switched sides in the 3 s before cue onset (posITI3: binary score, 1 indicates that the rat moved to the other side).

In order to validate our approach, we took the following steps. In the initial phase of our study, image analysis was performed in an iterative manner, i.e., the algorithm was retrained with images of rats that led to poor segmentation results. Training of the algorithm was stopped after results did not further improve, and no session in the study was excluded based on segmentation results. Segmentation quality was quantified based on visual inspection of the first 1000 frames of each session. First, we inspected whether the computed center-of-mass of a body part was aligned with the actual position in that frame. Error rates were low across rats (0.2 (0–0.8)%, $N = 70$ sessions) and therefore not routinely monitored for all sessions. However, the actual head direction was compared against the computed line connecting head-to-body for all sessions because error rates were higher: 8.2 (5.4–11.9)% ($N = 1100$ sessions). As a further measure of quality control, we correlated the difference between cue onsets identified based on gray scale value thresholds at cue light pixel coordinates with times provided by the MedPC behavioral control software for each session. Sometimes, frames corresponding to cue onsets were incorrect when rats were partially obscuring the cue light, leading to decreased correlation values. After manual correction of these errors, correlation values indicated successful identification of frames for cue onsets (0.996 (0.989–0.999)).

## Reinforcement learning (RL) models

We used attention-based RL models following the approach of Leong and colleagues[11] to test whether our strategy-related attention measures are behaviorally relevant and indeed represent empirical markers for attention at choice and learning. We assumed that rats use eight strategies to probe four task features for relevance (see Fig. 2a). We compared four RL models that differ in how attention modulates animal behavior: either during choice alone (attention-at-choice/AC model), learning (or reward feedback) alone (attention-at-learning/AL model), or both (ACL). This also allowed us to compare our data against the null hypothesis that attention modulates neither choice nor learning (uniform-attention/UA model). For attention-modulated RL models, we used both empirical attention scores measured by head direction and binary attention scores. We made a further comparison between attention-based RL models and standard RL models where rats learn all possible mappings between a multidimensional state space and actions. See Supplementary Methods for further details.

## Rat electrophysiology: data preprocessing

Raw data were first bandpass-filtered between 600–6000 Hz (Butterworth filter using MATLAB function *filtfilt*), and at each time point, the median across all channels was subtracted to reduce noise and remove artifacts[61]. Preprocessed data were then automatically spike sorted with Klusta (https://github.com/kwikteam/klusta) and afterwards manually curated with Klustaviewa (https://github.com/klusta-team/klustaviewa)[62]. See Supplementary Methods for details.

## Rat electrophysiology: decoding analyses

Decoding analyses of multiple single-unit data were performed by adapting MATLAB code from the Neural Decoding Toolbox (www.readout.info)[63]. Analyses were run using neural data from individual experimental sessions (using recordings from both the multidimensional rule-learning task and the probabilistic two-rule set shift). We included all sessions with ≥10 simultaneously recorded neurons and at least two different strategy types with ≥15 trials each (these numbers were based on toolbox recommendations[63] and the structure of our data). Based on these criteria, 62 experimental sessions from nine rats were included in all further analyses ($N = 4$ multidimensional rule-learning task, $N = 5$ probabilistic conventional set shift). Raw spike trains were first aligned to trial onset for each prefrontal neuron. Trial-aligned spike trains from trials of a behavioral category were concatenated for each neuron and labeled accordingly. Spikes were then binned using a sliding window approach (bin width 3000 ms, step size 300 ms). We used $z$-score normalized data to avoid that neurons with higher firing rates have a larger influence on the classification results. We used a maximum-correlation-coefficient classifier for decoding analyses based on leave-one-out cross-validation. The classifier learns a mean neural population vector for each class (i.e., a template for a behavioral category) by averaging all training points within each class. The classifier predicts the correct class using Pearson's correlation coefficient between the test data and the training set of each class (the highest correlation value corresponds to the predicted label). At least 15 trials per behavioral category were required: neural data from 14 trials were used as a training set to predict behavior in the test trial. Decoding accuracy for each session was based on 50 cross-validation runs using a resampling procedure (i.e., 15 trials per category were randomly drawn from the pool of all available trials per behavioral category in each run)[63]. See Supplementary Methods for more details.

## Human MEG: data preprocessing

MEG data preprocessing involved the segmentation of epochs of 2 s before to 4 s after stimulus presentation. Eye-movement-related activity and cardiac signals were identified with independent component analysis[64] and then discarded. All data were analyzed using the MATLAB-based toolbox for neuroelectric and neuromagnetic data analysis FieldTrip[65].

## Human MEG: source-level analysis

We computed forward models based on the MNI ICBM 2009 template brain (http://www.bic.mni.mcgill.ca/ServicesAtlases/ICBM152NLin2009). A parcellation scheme based on the Desikan-Kiliani atlas[66] was used. Following procedures similar to those described in Schoffelen et al.[67], single-dipole-specific spatial filters were concatenated across vertices comprising a given parcel. For each parcel, singular value decomposition was performed to extract spatially orthogonal and temporally uncorrelated components that describe the time course of activity within a parcel. These components were ordered by the amount of variance explained. Subsequently, the first principal component was selected, capturing the parcel's time course of activity. This procedure yielded 68 virtual sensors corresponding to each parcel that were subsequently treated in a similar fashion as MEG sensor-level activity.

Note that source-level analyses were also included because they increase the SNR for two reasons. First, sensor-level data represent a

linear superposition of all source activity plus the muscular, cardiac, and ambient noise, which potentially confound further analyses. In contrast, spatial filtering algorithms scan the entire source space voxel by voxel and treat the remainder of the signal, including all brain activity, as noise. Moreover, head movement within the MEG helmet introduces a variance on sensor-level data. Source analysis mitigates this confound and allows reliable estimates of the data at both descriptive and inferential levels.

## Human MEG: decoding analyses

Trial-based decoding was performed at the single-subject level using the MVPA-Light toolbox (https://github.com/treder/MVPA-Light)[68]. We decided to decode the current attentional focus (see Fig. 2a for a definition) instead of individual strategies in our primary analysis (e.g., during *go-light* and *go-dark*, subjects focus on the same task feature) to be able to include all available data sets in a random-effects group-level statistics (i.e., not all subjects displayed the same behavioral strategies). Since the strategy *win-stay-lose-shift* (task feature: outcome history) was not consistently detected across subjects (14/29 subjects eligible for MEG analysis), decoding analyses focused on the three remaining task features (auditory, visual, choice history; detected in 28/29 subjects eligible for MEG analysis) in 28 subjects. Moreover, we also used a complementary fixed effects analysis to test whether individual strategies can also be decoded in humans (see Supplementary Fig. 10 for details). Classification was performed using linear discriminant analysis, implementing 10-fold cross-validation with two repetitions. This means that the following process was performed twice: the entire data set was split into ten parts, with nine parts serving as a training set and one part as a test set. This was repeated ten times, such that each part served as a test set once. Thus, the number of trials used for decoding depended on the total number of available trials per category (i.e., it was not fixed as in rats). We computed discriminative information (expressed as the mean ± SEM decoding accuracy across subjects) across time to test whether decoding accuracy depends on sensory stimuli. We assumed significant decoding if the lower confidence interval of the curve exceeded the chance level for that time point. Moreover, we also compared decoding accuracy following stimulus presentation to the pre-stimulus baseline using a cluster-based permutation test[69]. A searchlight approach was used to highlight the source-level topography of decoding accuracy for the average of a selected time window. For one topographical visualization (Fig. 5k), we computed a within-subject contrast (paired *t*-test) between the baseline period (−500 to 0 ms) and the task period (0–1000 ms). This contrast was chosen because the null hypothesis that decoding performance during baseline and task stemmed from the same distribution was already rejected (Fig. 5j). To quantify the effect size of the within-subject contrast (task vs. baseline), we converted the resulting *t*-values to Cohen's *d* using the following formula: $d = t/sqrt(n)$, where *d* is Cohen's *d* for within-subject designs (also referred to as repeated-measures or paired-sample *d*), t is the *t*-statistic from the paired *t*-test, *n* is the number of participants[70].

## Statistics

Data analysis was performed with Graphpad Prism (Version 7), IBM SPSS Statistics (Version 29.0.0.0, IBM, Armonk, NY, USA), and MATLAB (R2017a or higher, MATLAB, Natick, MA, USA). All linear regression models were estimated with the MATLAB routine *fitlm*. To reduce the effect of potential outliers on model estimation, robust linear regression implementing the iteratively reweighted least squares algorithm was used. Normality distribution assumption was tested using the Shapiro-Wilk test. If systematic deviations from normality were detected, non-parametric statistical testing was used. Data are displayed as mean ± standard error of the mean (SEM) or as median and first and third quartile (Q_1–Q_3); Tukey-style whiskers were used in box plots.

## Reporting summary

Further information on research design is available in the Nature Portfolio Reporting Summary linked to this article.

## Data availability

Rat and human data sets generated in this study have been deposited in the Zenodo database under accession code: https://doi.org/10.5281/zenodo.15466504[71]. One human participant did not give consent to share data. Source data are provided with this paper.

## Code availability

Custom MATLAB code used for this study is available at https://doi.org/10.5281/zenodo.15466504[71].

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

## Acknowledgements

We thank the lab of Prof. Hamprecht for help with video analysis, Dr. Enkel for support during the development of the paradigm, and Prof. Draguhn for discussions. Some of the analyses required the use of a high-performance computing cluster (bwUniCluster 2.0, https://www.bwhpc.de/). The authors acknowledge support by the state of Baden-Württemberg through bwHPC. Figures include color specifications and designs developed by Cynthia Brewer (http://colorbrewer.org/). This work was funded by the German Research Foundation (BA 5382/1-1, BA 5382/2-2 to F.B., DU 354/8-2, DU 354/14-1 within FOR-5159 on "Resolving prefrontal flexibility," DU 354/15-1 to D.D.), the Schweizerischer Nationalfonds zur Förderung der Wissenschaftlichen Forschung (105314_207580 to T.P.), the third CIMH Young Investigator Award and the MACS (Mannheim Advanced Clinician Scientist) Program (F.B.).

## Author contributions

Experiments were designed by F.B. and T.P. and performed by F.B., T.P., S.H., N.B., T.M., and H.Z. Methods were developed by F.B., G.K., H.T., and D.D. Data were analyzed by F.B., T.P., S.H., N.B., T.M., H.Z., and H.T. Results were visualized by F.B., N.B., and H.T. This work was performed under the supervision of F.B., T.P., A.M.L., H.T., and D.D. The manuscript was written by F.B., T.P., and H.T. and edited by F.B., T.P., G.K., A.M.L., H.T., and D.D. All authors discussed the results and contributed to the finalization of the manuscript.

## Competing interests

The authors declare the following competing interests: A.M.-L. has received consultant fees from Daimler und Benz Stiftung, EPFL Brain Mind Institute, Fondation FondaMental, Hector Stiftung II, Invisio, Janssen-Cilag GmbH, Lundbeck A/S, Lundbeckfonden, Lundbeck Int. Neuroscience Foundation, Neurotorium, MedinCell, The LOOP Zürich, University Medical Center Utrecht, University of Washington, Verein für Mentales Wohlbefinden, von Behring-Röntgen-Stiftung; speaker fees from Ärztekammer Nordrhein, Caritas, Clarivate, Dt. Gesellschaft für Neurowissenschaftliche Begutachtung, Gentner Verlag, Landesärztekammer Baden-Württemberg, LWL Bochum, Northwell Health, Ruhr University Bochum, Penn State University, Society of Biological Psychiatry, University Prague, Vitos Klinik Rheingau; and editorial and/or author fees from American Association for the Advancement of Science, ECNP, Servier Int., Thieme Verlag. All other authors declare that they have no competing interests.
