## [Transparent Peer Review file · Nature Communications]

Abstract rule learning promotes cognitive flexibility in complex environments across species

Corresponding Author: Dr Florian Baehner

Version 0:

Reviewer comments:

Reviewer #1

(Remarks to the Author)

This paper asks how abstract rules are learnt in changing environments, and avails of approaches including empirical behavioral and neural data from rodents and humans. A key argument is that standard reinforcement learning does not easily explain flexible behavior – for example, performance transitions, rapid learning or faster learning with prior experience. The authors speculate that instead subjects learn hypothetical rules. To address they developed a novel “translational rule learning” task to ask whether “common computational mechanism exist, in humans and rodents”.

As a general comment the authors make a claim for species-conserved mechanisms in the title – this relates to rodent and human data. The argument for conserved mechanisms is not very convincing – what precise mechanisms are conserved? What data provides the key evidence for this claim? Given the very different neural data extracted from rodents and humans what are the precise features that make the case for conserved mechanisms?

The authors infer that both species follow low-dimensional behavioral strategies, testing each task feature for relevance, and go on to provide evidence that strategy related quantities can be decoded from a neural activity (single unit recordings in rat and MEG recordings in humans). A key claim is that that animals identify rules by “sequential testing different low-dimensional behavioral strategies”. In key behavioral findings the authors provide evidence for step change transitions, findings that challenge assumptions of incremental learning (as proposed in classic RL).

If I understand correctly the suggestion of sequential hypothesis testing predicts that there should be sensitivity to the number of potential strategies available. In other words, testing sequentially will be slower or faster depending upon the number of available strategies? Are there behavioral findings to support this?

The authors invoke selective attention to different task features as a mechanism for guiding behavioral responses. To formally test the role of attention they applied machine learning to video tape sessions from another group of rodents where they report onset and offset trials of each strategy were tethered to change in head direction of cue onset, towards and away from the task feature relevant for that particular strategy. Based on this the authors report strategy-specific changes in attention corresponding to a task feature, present for all eight strategies. This is in addition to a response that reward feedback which they term “attention at reward”.

I found it difficult to evaluate how precisely the machine learning approach was actually applied to identify strategy specific changes in attention – this needs to be described much clearer, including validation of the approach.

Fitting measures of attention to a range of models they found that a model that includes attention at choice and attention at learning (ACL) provided the best fit including being best at predicting held-out behavioral data. The authors go on to show new strategies are learnt if behavioral work controls do not match current task demands. Moreover, the value-based selection of established strategies provides an important selection mechanism that explains transfer effects.

The notion of transfer effects are important but this aspect of the manuscript is unclear – the implication is that value-based strategies alone act as an action selection mechanism in mediating transfer effects? Have they examined in detail evidence for other transfer effects?

In neural data the authors explore abstract representation of strategies and task features in rodents prefrontal cortex which they claim represents high-level behavioral strategies consistent with an abstract rule that is independent of the particular motor action. Here this inference relies on decoding from rat pre-limbic area. Here the authors report that decoding accuracy is already above chance prior cue onset.

In relation to this, Figure (4a) shows that decoding accuracy pre-cue onset is as high as that seen after cue onset; and is reported not to differ between conditions. This is a surprising finding. Can they provide a clearer account of why this should be the case? Would the authors have not expected a significant increment of decoding accuracy following onset of a cue?

The authors report accuracy exceeds chance over the 3 seconds prior to cue onset; this raises a question related to the dynamics of this response – when does decoding accuracy first exceeds chance prior to cue onset? Could they provide more detail and data on this? Can they provide an account of what precisely triggers the significant increment in decoding accuracy pre-cue (at the time this increment is first observed)?

In a final experiment they test data from a human MEG, using a modified task, where behavioral analysis is said to be consistent with that seen in rodents. In MEG decoding they focus on task feature using a search light analysis at the source level.

A problem here is that the human data is sparsely described by comparison to the rodent data. This leads to a number of concerns not least that it is difficult to know precisely the reasoning strategy deployed in relation to the MEG analysis.

Firstly, it is not at all clear why the analysis was carried out at source level, given that this is usually a much noisier metric than sensor level analyses. Did they authors examine sensor level data? The decoding accuracy plots fluctuate over time but an apparent peak at roughly 0.75 seconds (as opposed to a more sustained accuracy seen in the rodent data). Can the authors address this apparent discrepancy?

As reported in their rodent data there is reference to decoding accuracy being above chance prior to cue onset. However, the data seem to show an increment in accuracy at cue onset – is this a significant increment. The figure related to the MEG data itself is not very informative (two small panels). The relevant figures are not well prepared – for example, the authors should indicate in the figure 5i what chance level (perhaps it is there but poorly reproduced).

It is not entirely clear how decoding accuracy was tested prior to implementation of this analysis? How was it tested? What precisely is a “task feature tested for relevance by a strategy”? I doubt this will be understood by the readership.

(Remarks on code availability)

Reviewer #2

(Remarks to the Author)

Summary

This paper provides an elegant comparison of rule learning strategies across rats and humans. The authors show that rather than inefficient state-to-action learning in high-dimensional environments, learning performance in rats and humans indicates that both species reduce the dimensionality of the task space, thereby learning a low dimensional representation that allows for efficient decision-making despite changes to task rules. In addition, the authors show that by modelling attention during choice, they can explain sudden transitions in behavior to select and pursue a particular behavioral strategy. The authors implement cross-validated model comparison to demonstrate that attention at choice provides the winning model for behavior on their task. In addition, the authors show that behavioral strategies can be decoded from PFC in both species, using single-unit recordings in rats, and MEG data in humans.

Overview

This type of cross-species work is rare and challenging. I commend the authors for acquiring data in both humans and rats, using a comparable and non-trivial task. While previous studies have shown evidence for abstract rules being represented in PFC, here the authors provide insight into how attention may play a critical role in guiding behavior that draws from a low-dimensional representation. The behavioral analysis is careful and informative. The neural analysis could potentially be elaborated to strengthen the paper. Below are a few comments and suggestions to consider.

Comments

- Training (a clarification point): It is not clear how much training the rats received – when you say ‘rats were not pre-trained on any rule’, does this mean that they received no training at all, or they received training on other rules, or different orders of rules? Please can the authors make this clearer, in the Results section in particular. Related to this point, how do the authors account for an order effect? Is the order of task rules counterbalanced across subjects, or kept consistent?

- Cognitive process (a clarification point): if subjects are not pretrained, can the authors be clearer still on how the representation of the eight low-dimensional behavioral strategies are learned? The authors state that ‘many rats did not use the correct strategy before it was required’, and ‘rats learn new strategies if the behavioral repertoire does not match current task demands’. Does the behavioral modelling therefore show that different strategies are acquired according to the

experimenter-defined order of task rules, such that some are represented early on, while others are only represented later on in the task? If this is the case, then can we really consider the rats to be performing hypothesis testing across all eight different behavioral strategies, if they only acquire an eight dimensional representation at the end of the experiment?

- What neural computation is being modeled here? To perform this task, subjects need a representation of the low-dimensional state space, and a means to sequentially test and evaluate different options, before selecting and opting for a particular strategy. In other words, there are multiple processes at play here. Can the authors more clearly articulate which neural processes they are trying to account for in PFC? Are we observing changes in sampling of an established state space, or is the state space itself is represented differently as the task proceeds?

Mouse data:

- Hypothesis testing: For Fig 4d and 4g: the significant decoding of behavioral strategy independent of current action is reported during cue and following lever onset. What does this mean in terms of neural computation? Would the authors suggest that a brain region other than PFC is performing hypothesis testing prior to choice?

- Differences in PFC subregion: Were there differences in the neural code between Cg1, PrL, IL? What is the average response of these regions to the task? As a sanity check, it would be reassuring to see some more standard physiological analyses for ephys data acquired from these regions.

Human data:

- Classifier: Is it not possible to train and test on the strategies that are included for a given subject? The decoding of cue prior to onset shows predictive activity but doesn't necessarily show evidence for coding of strategies.

Significance of the effect: Fig5h: The distribution of the data across subjects is not shown. Please show the distribution of individual data points and make clear how significance is determined. Given that large areas of the brain appear to show significance it isn't clear that this is an effect specific to PFC. Moreover, it isn't clear which time bin this data comes from and how the time bin was selected (with multiple comparisons in mind). Is this prior to cue onset as suggested to be the case in rats? Similarly, Fig 5i: how is significance assessed here? Please indicate the chance level on the plot. Is this using decoding of relevant features, strategies, or something else?

Cross-species comparisons:

- Task design: From Fig 5a it is hard to understand what the human task involves and how it provides a direct comparison to the one used in rats. Can the authors provide a more detailed schematic and highlight points of comparison across species?

- Learning curves: While the authors show the learning curves for models, they do not show the equivalent learning curve for raw behavioral data from rats (except for Fig 1d) and humans. Please can the authors add this in to help provide qualitative comparison of the step changes in subject learning vs ACL model learning?

- Analyses: Can the authors provide more direct comparisons across species? For example, does the reaction time effect in humans replicate in rodents? Does the multiple regression analysis applied to rodent behavior pull out the same results in humans?

(Remarks on code availability)

Reviewer #3

(Remarks to the Author)

Species-conserved mechanisms of abstract rule learning promote cognitive flexibility in complex environments

Bähner, Popov, Boehme, Hermann, Merten, Zingone, Koppe, Meyer-Lindenberg, Toutounji, and Durstewitz

This manuscript investigates rapid learning in complex environments, a fundamental aspect of intelligent behavior. Humans are thought to use abstract strategies to focus on particular aspects of a task to speed learning (sometimes referred to as 'hypothesis testing'). However, it remains unknown if similar mechanisms exist in other species. To address this question, the authors used a combination of behavioral, computational, and electrophysiological analyses in both rats and humans to study how subjects learned new tasks. Rats performed a multidimensional task with four possible features guiding response: auditory stimulus, visual stimulus, reward history, and choice history. During each block of trials, subjects had to discover the relevant feature dimension through trial and error. Humans performed a similar task variant. Using statistical techniques and computational modeling of behavior, the authors found evidence that both species sequentially sampled different 'strategies', with a strategy defined as what task feature was being attended. In other words, the subjects were not learning the Q-value in the full dimensional feature space. Furthermore, by combining electrophysiological measurements in rats and magnetoencephalography (MEG) in humans with classifier analysis, they demonstrated that the prelimbic cortex in rats and the prefrontal cortex in humans encode abstract task strategies.

Overall, this is an interesting study with clever task design and rich dataset. The manuscript is clearly written with comprehensive review of relevant literature in the introduction. The results are exciting and consistent with a growing body of work that suggests learning does not necessarily occur at the level of raw features but can take advantage of lower-dimensional representations. However, despite our general enthusiasm, we do have some comments:

Major comments:

1- The authors' find rats used strategies even when the reward feedback was random. If true, this provides strong evidence that rats sample different strategies but also needs stronger verification. It would be helpful to include a figure depicting the behavioral performance of naïve and non-naïve rats with random reward feedback, along with an analysis demonstrating that these animals attend to stimuli if their detected strategy includes a stimulus.

It also isn't clear why naïve rats would employ any strategies. If the animal does not understand the levers lead to reward, then why would they push the levers at all? Were they also pretrained to push levers for reward? If so, then one might expect they engage in the random response, perseveration, and Win-Stay Lose-Shift (WSLS) strategies but it still isn't clear as to why they would respond to stimuli that were meaningless (even if they were familiar with the cues).

Furthermore, it isn't clear as to why the animals would have this basis set of strategies. These strategies don't seem particularly ecologically relevant given the artificial nature of the stimuli.

2- Related to the above comment, the ability to identify strategies should be more completely verified. For example, were the same strategies found in simulated data from a randomly responding model? Or simulations from a model that was using a few of the strategies (e.g., just win-stay-lose-shift or just a subset of the strategies). Or in the data from the model that was designed to sample from different strategies? Some quantification of the ability of the algorithm to reliably identify sequences of strategies would be helpful.

3- The authors assert that their defined strategies are lower-dimensional than the full space of Q-values. Isn't this only true because the authors selected a subset of strategies to test? Other strategies, such as arbitrary associations of light position, sound, and previous response, would greatly expand the space of strategies. How were these strategies chosen? Were all necessary to explain the data? Did expanding the set of models improve behavioral fits? To address these concerns, we would suggest the authors:

- Incorporate mixed strategies into their strategy detection algorithm and demonstrate that rats do not use these more complex strategies but prefer the low-dimensional eight strategies defined in the task.
- Demonstrate that the reduction in dimensionality is due to attentional mechanisms. Specifically, when rats employ a specific strategy, trial-by-trial behavioral choices should correlate with the attentional factors defined in their model (WAC and WAL). Conversely, trial-by-trial behavioral choices should weakly correlate with attentional factors when animals use mixed strategies.

4- It wasn't clear to us how the ACL model, where attention acted on both the choice and learning, differed from attention filtering of the stimulus input. This seems like the more common way of thinking about attention – that it filters sensory stimuli, allowing both learning and choice to be driven by the attended feature. If these models are not equivalent, then we would suggest fitting a new model where attention acts on sensory inputs to both choice and learning.

5- Related to the previous comment, were there any differences in the attentional modulation between species?

6- If rat prelimbic cortex represents the animal's abstract strategy, then was the strength of the strategy classifier (Fig. 4c) correlated with the attentional factors estimated by the model?

7- Based on Figure 1d, it seems that rats will occasionally adopt the correct strategy before abandoning it. Why would that occur? In the authors' model, once the correct strategy is identified the animal should persist, which is what leads to the rapid learning observed in the animal's behavior.

8- The authors use the animal's body orientation to help identify the strategy they were using. However, this raises the concern that neurons in prelimbic cortex are not encoding strategy per-se but rather the body position, etc. that is being driven by the strategy. For example, previous work has shown that neurons in the rat prelimbic cortex encode spatial locations (Hok et al., 2005, PNAS) and the author's show that the animal's head direction changed based on the strategy (Fig. 2a,b). Is there any way to exclude these potential confounds in Figure 4? Similar questions could be ask for human participants. If not, then perhaps the authors can discuss these concerns in the Discussion section.

9- Were classifiers trained on balanced number of trials of each experimental condition? In general, we would suggest including more detail in the description of the classifier in the Methods section.

10- The authors repeatedly claim that rats learned rapidly. However, they experience hundreds to thousands of trials before learning (e.g., Figure 1d). Why should we consider this to be rapid learning? How should we interpret this relative to the dozens of trials that humans require?

Related to this comment, it isn't clear as to why performance in naïve rats declines over the initial four days (Fig. 1d)? Even if sampling strategies randomly, shouldn't it be stable until they hit upon the right one?

11- How many strategy switches did the animal's experience before they stabilized on the correct strategy? How many times did they visit each strategy? Did they revisit poorly performing strategies?

12- On any given trial, there may be multiple feature changes that could lead to a change in orientation (such as seen in Figure 2a). Can the effect of different features be dissociated?

13- It isn't clear how well the models fit the behavioral data. Figure 2d shows the ACL model fits better than others, but the overall quality of fit isn't clear. It would be useful to include the animal's behavior in Figure 2e-g for comparison.

14- In Figure 4d, 4f, and 4g – what is chance level? We assume its 50%? If so, then are all of these significantly greater than chance? While the differences between same and different are interesting, they can only be interpreted if at least one of them is greater than chance.

Minor comments:

1. What are the region names in Fig. 4a?

2. What is the range of performance along the y-axis of Figure 1d? It is important to clarify how much of the performance varies across time.

3. From Figure 1d, it seems as if strategies overlapped in time. Is this true? From our understanding of the approach, only a single strategy should be enacted at a time?

4. In Figure 2a, when comparing across switch point – were the stimuli and response matched across trials? If not, could the observed results reflect these other changes rather than just the internal change of strategy?

5. It would be helpful to indicate which horizontal lines in Fig. 2e-g represent the task switches and which indicate changes in the animal's strategy.

6. It would be helpful to clarify the meaning of "multi-rule rats" in the title of Fig. 2d.

7. The trial labels are missing on the x-axis of Figs. 2e-g.

8. In the legend of Figure 2c, what is "filled colored circles" referring to in line 235?

9. The "go-silent" is not italicized on line 270.

10. The y-axis of Figs. 3h and 3j are not labeled.

11. No statistics are provided to support the statement on line 345.

12. What regions in Fig.5h are significant? What timepoints in Fig. 5i?

13. Should "alternate" be within the parentheses of "win-stay-lose-shift" in the supplemental information, line 94?

14. In Fig. S1b, what was the criteria use to define an animal as a "bad-learner"?

(Remarks on code availability)

I didn't try to execute the code, but I looked through the readme files and everything seems to be clearly explained.

Reviewer #4

(Remarks to the Author)

This manuscript explores computational and neuronal underpinning of abstract rule learning in rats and human participants using behavioral, electrophysiological and MEG experiments.

Unfortunately, the data, the analyses, visualization and arguments are not sufficiently convincing to support the conclusions, at the level necessary for Nature Communications. There are several issues with the paper:

The analyses presented across the manuscript are not performed in sufficient depth, they are not clearly motivated and they are often not separating different hypotheses from each other. For example, the first experiment/figure concludes that "rats test low-dimensional behavioral strategies rather than learning high-dimensional mappings between environmental states and actions". It is unclear what analyses clearly establishes that rats use low-dimensional behavioral strategies. What in the experimental design and analyses separate/diagnose the use of low-dimensional behavioral strategies from that of high-dimensional behavioral strategies? How does this section/analyses leads to "selective attention" section (Fig.2)?

The presentation and analyses of electrophysiological experiment does not meet the norm standard in the electrophysiology field. Several studies have shown the complexity and challenges in analyzing PFC data during complex behavioral tasks. This study brushes over this, only showing a few rather indirect and abstract figures to describe the data and results.

The manuscript uses various forms of computational models without formally motivating them.

Technical phrases, often used in the field in specific ways, are utilized in conjunction with each other without sufficient clarity, making the manuscript very hard to follow. Two examples are the following: strategy-specific selective attention, and value-based selection of behavioral strategies.

Many figure panels have some visualization issues. For example, Fig.2 e,f and g the numbers are the x axis missing, making it hard to understand the figures.

(Remarks on code availability)

Version 1:

Reviewer comments:

Reviewer #1

(Remarks to the Author)

The authors have provided a careful response to the critiques of the four referees. In particular they have carefully addressed elements of the manuscript which were not entirely clear in the original submission. This has improved its accessibility.

I still struggle with the human data. The authors have expanded on points of clarification sought in my original review. The broad claim, as stated in the abstract, is that neural substrates of hypothetical rule were detected in pre-frontal activity in both rodents and humans. This entails making a claim for species conserved mechanisms, where this refers to reduced task dimensionality. This begs a huge question as to the “homology” between the rodent and human task version. Likewise, the claim that the findings narrow the explanatory gap between human macroscopic and rodent microcircuit levels is quite a claim, and not entirely convincing based on the data.

So my remaining concern is with the exposition of the MEG data and the claim this is homologous to mouse data!

(Remarks on code availability)

Reviewer #2

(Remarks to the Author)

The authors have responded to all my comments with extensive revisions to the manuscript. I support publication of the manuscript.

(Remarks on code availability)

Reviewer #3

(Remarks to the Author)

Species-conserved mechanisms of abstract rule learning promote cognitive flexibility in complex environments

Bähner, Popov, Boehme, Hermann, Merten, Zingone, Koppe, Meyer-Lindenberg, Toutounji, and Durstewitz

This is a re-review of this manuscript. As before, I feel this is an interesting manuscript that uses a series of complex behavioral and neural analyses to study strategy switching in both rats and humans. In my opinion, the manuscript is timely and will be of broad interest. The revisions strengthened an already strong manuscript and largely addressed my previous comments. I clarified a couple of our previous comments below, which I hope the authors would be able to address.

1. Our previous comment #4 noted that it wasn't clear to us how the ACL model differed from attention to sensory inputs. I apologize if our comment wasn't clearly communicated, but I'm not sure the authors response address this concern. In their response, the authors seem to first agree that the ACL model is equivalent to a sensory filtering model. But then they appear to address a slightly different question – whether attention is either binary or graded. Either binary or graded attention could be encapsulated by sensory filtering (I think the reviewers agree with this as well?). To restate the previous point – the literature tends to discuss attention as acting on sensory inputs. Filtering at an early stage would impact both choices and learning. Here, the equivalent model seems to be the ACL model where attention acts on both choice and learning. As noted before, these seem equivalent and I would suggest a) making this clear in the manuscript and b) adopting the simpler terminology/framework used in previous work, where attention modulates stimulus inputs. I would strongly recommend the authors at least do (a) but would leave (b) up to them and their style.

2. Our previous comment #8 noted that, given the fact that the animals orientation and location in space was correlated with

their behavioral strategy, the neural responses observed in prefrontal cortex of rats could reflect those behavioral variables rather than (or in addition to) encoding the behavioral strategy. In their response the authors address a potential motor confound. Does this also encapsulate the position differences (i.e., without the motor action itself?). Are head direction, body orientation, and location all perfectly correlated with the motor response? My impression from the manuscript was that this was not true, which means there could still be a correlation between orientation/location and strategy even after controlling for motor responses. In this case, our question remains – can these be fully dissociated? If not, I would suggest noting this potential confound in the Discussion.

3. Our previous comment #3 was asking whether removing strategies from the model fits reduced the accuracy of the overall fit – that is, were the animals always using all possible strategies or could some of them be removed from the fit? The authors response was not clear – I understand that the rat would eventually be trained on tasks that required the use of all strategies, but this doesn't mean that the animal was always using or sampling all of the strategies throughout the process. Similarly, one could imagine adding new strategies to the mix, even if they were never tested in a task, and improving the model's fit to the animals behavior (this was the intention behind what the authors designated as comment 3c; we incorrectly wrote 'models' instead of 'strategies' which probably led to the confusion).

(Remarks on code availability)

Reviewer #4

(Remarks to the Author)

The manuscript has improved in presenting the behavioral data.

The rat mPFC results are still not convincing. I appreciate that authors have attempted to show less processed neuronal data. However, what is shown as an 'example neuron' is arguably an outlier, as evident in the histogram of fig 4c. Moreover, it remains unclear how neurons respond to task events. In past studies a significant proportion of mPFC neurons respond to events (cues, lever presses) in a task with transient changes in the firing rate, and this does not happen to be the case here, as shown in fig 4c. Given this, the quality of spiking data is unclear to me, and the example neuron shown suggest to me that the recordings are not sufficiently stable.

Figures have improved given my previous comments. However, some labels are still missing.

(Remarks on code availability)

NA

Version 2:

Reviewer comments:

Reviewer #1

(Remarks to the Author)

My concerns related to homology of MEG and rodent task remain. I still consider that the linkage is weak notwithstanding the abbreviated presentation of the MEG findings! At the very least I would strongly advocate that the manuscript title is changed - given the controversial issue around this particular issue

(Remarks on code availability)

Reviewer #3

(Remarks to the Author)

The authors have addressed my comments. Congratulations on a very interesting paper!

(Remarks on code availability)

I've reviewed it in previous rounds of revisions and it seemed sufficient, although I did not try to execute it myself.

Reviewer #4

(Remarks to the Author)

(Remarks on code availability)

REVIEWER COMMENTS

Thank you for the thoughtful comments, which led to multiple additional analyses that helped to strengthen the main claims of this manuscript. We also removed some material from the manuscript (previous Supplementary Figures 1, 5 and Supplementary Table 3) due to redundancies after addressing all comments. We numbered all reviewer comments to facilitate point-by-point responses.

Reviewer #1 (Remarks to the Author):

This paper asks how abstract rules are learnt in changing environments, and avails of approaches including empirical behavioral and neural data from rodents and humans. A key argument is that standard reinforcement learning does not easily explain flexible behavior – for example, performance transitions, rapid learning or faster learning with prior experience. The authors speculate that instead subjects learn hypothetical rules. To address this they developed a novel “translational rule learning” task to ask whether “common computational mechanism exist, in humans and rodents”.

1. As a general comment the authors make a claim for species-conserved mechanisms in the title – this relates to rodent and human data. The argument for conserved mechanisms is not very convincing – what precise mechanisms are conserved? What data provides the key evidence for this claim? Given the very different neural data extracted from rodents and humans what are the precise features that make the case for conserved mechanisms?

Thank you for raising this important and complex point. Our study aims to show that some of the computational mechanisms underlying rapid learning in complex environments are species-conserved. Our core hypothesis is that learning and testing low-dimensional hypothetical rules (“behavioral strategies”) to infer task rules is a key mechanism used by both rats and humans. This hypothesis was tested as follows: first, we developed a strategy detection algorithm along with behavioral modeling to show that several characteristics of response behavior such as the observed learning variance and the presence of step changes in performance can best be explained by the use of behavioral strategies in both species. Second, the existence of behavioral strategies in rats and humans is well-documented in the literature (see Introduction p. 4) and it has been speculated that it could be a useful heuristic (e.g., in the sense of an exploratory behavior) but the precise role in higher-order cognition was less clear. We aimed to arrive at a more detailed understanding of the computational processes during hypothesis testing by using RL models and testing their predictions on behavioral and neural data. More specifically, we used attention-modulated RL models in both species to show that selective attention organizes action selection and learning when individuals test different hypothetical rules. In order to understand how a strategy along with a corresponding attention focus is selected we hypothesized that there is an additional higher level for action selection and learning at the level of strategies. Such a hierarchical mechanism was tested indirectly based on experimental predictions of hierarchical RL models. These included neural decoding analyses in both species and a characterization of transfer effects in rats.

During the revision process, we added several additional analyses to strengthen translational links:

- We added several analyses to further validate the results of the strategy detection algorithm in rats and humans. Both naive humans and rats performing a task version

with random reward feedback did not behave like randomly choosing agents. The percentage of trials that can be explained by strategies was significantly higher for naive subjects performing the random rule task (i.e., the detected strategies are not merely false-positives; Fig. 1c, 5b). Moreover, head-direction plots in rats (Supplementary Figs. 2-4) indicate that even in the context of random reinforcement, choices are not random but appear to be goal-directed (i.e., they can be predicted from head movements that occur 3s before lever onset). In order to evaluate the false-negative rate, we used synthetic strategy sequences of varying lengths to evaluate the percentage of trials that was correctly detected by the strategy detection algorithm. A statistical power of >80% is reached for sequence lengths of ≥ 9 trials. These results are shown in Fig. 1c. See also comment #2 from reviewer 3 for further discussions on validation of the algorithm.

- RL models showed that rats and humans do not learn the full state-action-space or variants thereof but potential alternative explanations for the observed response patterns remained and were highlighted in reviewer comments (reviewer 3+4). This is a very important point we had not made clear enough. We removed panel 1b because it suggested that learning the full state-action-space is the only alternative to the proposed strategies. Rather, individuals could also use alternative strategies that are suitable for rule learning and this would be compatible with performance steps. Indeed, using a win stay-lose shift vs. win shift-lose stay approach with respect to place, visual and auditory cues to learn rules would fulfill these criteria and be in line with the literature on the ecological validity of such strategies (Evenden JL & Robbins TW, *The Quarterly Journal of Experimental Psychology B* 1984; Dember WN & Fowler H, *Psychol Bull* 1958; Maggi S et al., *Elife* 2024). While only six strategies would be required to learn all rules, these alternative strategies would be higher-dimensional than the proposed set of strategies because they always depend on two task features. We adapted the strategy detection algorithm and again counted the percentage of trials explained by alternative strategies in individuals performing the random rule versus a random agent. In both species, behavior was not different from a random agent which indicates that neither rats nor humans make extensive use of such strategies (see Figs. 1c, 5b).
- We fitted another set of attention-modulated models to the rodent data, but instead of using attention scores measured by head direction, we used binary attention scores as in the models fitted to human data (i.e., the rat is attending fully to the current task cue). There was no difference between the ACL and binary ACL model which both explained the data best as compared to all other models. This indicates that attention is a very strong filter and is thus evidence for low-dimensional strategies in both species (Figs. 2d, 5e).
- We also added electrophysiological analyses in rats and humans to make them more comparable. More specifically, we now show that the current attention focus can not only be decoded from human MEG but also from rat prelimbic recordings and that it is also possible to decode individual strategies from human MEG using a fixed-effects analysis (Fig. 4e, Supplementary Fig. 9c,d).

We do agree that neural data from rats and humans have different characteristics with complementary strengths. However, a comparison of neural analyses across species can still provide valuable insights if the translational paradigm engages similar cognitive processes (Nour MM et al., *Neuron* 2022; Barron HC et al., *Philos Trans R Soc Lond B Biol Sci* 2021). If the proposed computational processes can be decoded from neural signals of similar brain regions at different spatial scales, it provides evidence that behavioral models are relevant for brain computations. This is reminiscent of findings on neural correlates of RL quantities such as action values or reward prediction errors in human fMRI or animal single-unit recordings

that have stimulated further theoretical and experimental work in multiple species (Schultz W et al., *Curr Opin Neurobiol* 2017; Lee D et al., *Annu Rev Neurosci* 2012).

We restructured the discussion (pp. 27-31) accordingly and added a schematic of computational processes underlying task rule inference (Fig. 6).

The authors infer that both species follow low-dimensional behavioral strategies, testing each task feature for relevance, and go on to provide evidence that strategy related quantities can be decoded from a neural activity (single unit recordings in rat and MEG recordings in humans). A key claim is that that animals identify rules by “sequential testing different low-dimensional behavioral strategies”. In key behavioral findings the authors provide evidence for step change transitions, findings that challenge assumptions of incremental learning (as proposed in classic RL).

2. If I understand correctly the suggestion of sequential hypothesis testing predicts that there should be sensitivity to the number of potential strategies available. In other words, testing sequentially will be slower or faster depending upon the number of available strategies? Are there behavioral findings to support this?

Thank you for your comment. We compared how fast naive rats acquire the rule go-light either in the multidimensional rule learning task or a conventional operant rule learning paradigm described in Supplementary Fig. 1 (Floresco SB et al., *Behav Brain Res* 2008). Indeed, rats in the multidimensional task were significantly slower to reach the learning criterion (84.6 +/- 18.1 vs. 1064 +/- 105.4 trials, $p=1.96*10^{-9}$, unpaired t-test). Yet, once the learning criterion was reached, the length of *go-light* sequences detected were not different (46.5 (35.3-68.8) vs. 37 (25-57), $p=0.33$, Mann-Whitney test), which indicates similar performance in both groups (now shown in Supplementary Fig. 1c). We refer to these findings in the Results (p.8).

The authors invoke selective attention to different task features as a mechanism for guiding behavioral responses. To formally test the role of attention they applied machine learning to video tape sessions from another group of rodents where they report onset and offset trials of each strategy were tethered to change in head direction of cue onset, towards and away from the task feature relevant for that particular strategy. Based on this the authors report strategy-specific changes in attention corresponding to a task feature, present for all eight strategies. This is in addition to a response that reward feedback which they term “attention at reward”.

3. I found it difficult to evaluate how precisely the machine learning approach was actually applied to identify strategy specific changes in attention – this needs to be described much clearer, including validation of the approach.

Thank you for pointing this out.

We used the open source software Ilastik for machine learning-based image analysis (Ilastik-1.2.0, <http://ilastik.org/>). Frames were segmented into the background, head and body of a rat. In order to validate our approach, we took the following steps. In the initial phase of our study, image analysis was performed in an iterative manner, i.e., the algorithm was retrained with images of rats that led to poor segmentation results. Training of the algorithm was stopped after results did not further improve and no session in the main study needed to be excluded based

on segmentation results. Segmentation quality was quantified based on visual inspection of the first 1000 frames of each session. First, we inspected whether the computed center-of-mass of a body part was aligned with the actual position in that frame. Error rates were low across rats (0.2 (0-0.8) %, N=70 sessions) and therefore not routinely monitored for all sessions. However, the actual head direction was compared against the computed line connecting head-to-body for all sessions because error rates were higher: 8.2 (5.4-11.9) % (N=1100 sessions). As a further measure of quality control, we correlated the difference between cue onsets identified based on grey scale value thresholds at cue light pixel coordinates with times provided by the Med-PC behavioral control software for each session. Sometimes, frames corresponding to cue onsets were incorrect when rats were partially obscuring the cue light, leading to decreased correlation values. After manual correction of these errors, correlation values indicated successful identification of frames for cue onsets (0.996 (0.989-0.999)).

We added this information to the respective Methods part (pp. 41-42).

Fitting measures of attention to a range of models they found that a model that includes attention at choice and attention at learning (ACL) provided the best fit including being best at predicting held-out behavioral data. The authors go on to show new strategies are learnt if behavioral work controls do not match current task demands. Moreover, the value-based selection of established strategies provides an important selection mechanism that explains transfer effects.

4. The notion of transfer effects are important but this aspect of the manuscript is unclear – the implication is that value-based strategies alone act as an action selection mechanism in mediating transfer effects? Have they examined in detail evidence for other transfer effects?

We agree that this point needs clarification. It is correct that only some of the observed transfer effects can be explained by value-based selection of established strategies (Fig. 3). We decided to include details on strategy formation (which are not central for understanding transfer effects) in a new Supplementary Fig. 7. We also restructured Fig.3 such that it now focusses exclusively on transfer effects and instead added the following analyses to main Figure 3d.

Given that attentional mechanisms are relevant for rule learning in both groups (Fig. 3b, c), we tested whether faster learning in experienced rats can be related to more focused attention. More specifically, we estimated a regression model (N=89 rats, $R^2_{\text{adjusted}}=33.3\%$) and found that the learning trial can be predicted by median attention-at-choice (for all trials preceding the learning trial; $p=1.6 \cdot 10^{-5}$) but not median attention-at-reward ($p=0.86$). A variable coding for learning experience was also significant ($p=0.004$) which indicates that experienced rats are more focused than naïve rats even when they require the same number of trials to learn the task rule). Average angles at cue onset were lower in experienced (N=51) vs naïve (N=38) rats: 10.4 (9-13.8) vs. 13.2 (11.1-16.3) $^\circ$ ($p=6 \cdot 10^{-4}$, Mann-Whitney test), indicating higher levels of attention in experienced rats.

In neural data the authors explore abstract representation of strategies and task features in rodent prefrontal cortex which they claim represents high-level behavioral strategies consistent with an abstract rule that is independent of the particular motor action. Here this inference relies on decoding from rat pre-limbic area. Here the authors report that decoding accuracy is already above chance prior cue onset.

5. In relation to this, Figure (4a) shows that decoding accuracy pre-cue onset is as high as that seen after cue onset; and is reported not to differ between conditions. This is a surprising

finding. Can they provide a clearer account of why this should be the case? Would the authors have not expected a significant increment of decoding accuracy following onset of a cue?

The fact that decoding accuracy of a strategy is above chance during the inter-trial interval is further evidence that the prefrontal cortex is encoding the hypothetical, abstract rule the rat is currently following rather than low-level cues and actions. This is in agreement with previous studies using multiple single-unit recordings in rats and primates that showed that prefrontal coding of a rule is a global feature of its dynamics, and is not locked to any individual task-related events like cue onset or lever press (Durstewitz et. al., *Neuron* 2010; Rodgers CC et al., *Neuron* 2014; Mansouri FA et al., *J Neurosci* 2006). We added new material to make this more salient. We present an example of strategy coding throughout trial phases at the level of an individual unit (new Fig. 4c) similar to what has been observed for prefrontal rule representation (in the steady state) in the references cited above. Moreover, we compared strategy decoding at the population level with decoding of cue location and side of lever press which is not prominent prior to cue onset (new material added to Fig. 4e).

6. The authors report accuracy exceeds chance over the 3 seconds prior to cue onset; this raises a question related to the dynamics of this response – when does decoding accuracy first exceeds chance prior to cue onset? Could they provide more detail and data on this? Can they provide an account of what precisely triggers the significant increment in decoding accuracy pre-cue (at the time this increment is first observed)?

We conducted a population analysis for the entire 20s inter-trial interval (bin width 3000 ms, step size 300ms) and found that the minimum value for decoding accuracy was $64.3 \pm 1.2\%$ (N=105 strategy pairs) which occurred 10.2 (6.68 - 14.1) s prior to cue onset. This was significantly lower than for the three seconds prior to cue onset (1.1×10^{-25} , paired t-test) but higher than the 50% chance level (8.9×10^{-21} , one-sample t-test). As stated above, we argue that the action selection during strategies is a top-down process and prefrontal representations of strategies are therefore not expected to be strictly cue-dependent. We included this information in the Results (p. 19).

In a final experiment they test data from a human MEG, using a modified task, where behavioral analysis is said to be consistent with that seen in rodents. In MEG decoding they focus on task feature using a search light analysis at the source level.

7. A problem here is that the human data is sparsely described by comparison to the rodent data. This leads to a number of concerns, not least that it is difficult to know precisely the reasoning strategy deployed in relation to the MEG analysis.

We expanded the Methods section on MEG analysis (pp. 45-47) and added a schematic for human task design to Fig. 5a. We also added new behavioral analyses to show that humans actually employ low-dimensional behavioral strategies (new Fig. 5b). Moreover, we link performance in our task to established markers of executive function as measured using the CANTAB neuropsychological test battery that all subjects completed (Supplementary Table 5). At the neural level, we now provide a sensor-level analysis of the results shown in Fig. 5k (Supplementary Fig. 9b; see also next comment) and a source-level fixed-effects analysis based on decoding of individual strategies (Supplementary Fig. 9c, d).

8. *Firstly, it is not at all clear why the analysis was carried out at source level, given that this is usually a much noisier metric than sensor level analyses. Did the authors examine sensor level data? The decoding accuracy plots fluctuate over time but an apparent peak at roughly 0.75 seconds (as opposed to a more sustained accuracy seen in the rodent data). Can the authors address this apparent discrepancy?*

We thank the reviewer for this important note. A sensor-level analysis was added to Supplementary Fig. 9b. However, there is a slight misunderstanding with respect to source- and sensor level-analyses. More specifically, source space analysis, particularly using spatial filtering algorithms as in the present case, scan the entire source space voxel by voxel and treats the remainder of the signal, including all brain activity, as noise. An adaptive spatial filter is created that effectively renders the data with a higher signal-to-noise ratio (SNR) compared to the sensor level data, where the linear superposition of all source activity plus the muscular, cardiac, and ambient noise are mixed and potentially confound further analyses. Unlike EEG, head movement within the MEG helmet introduces a variance on sensor level data that should be accounted for. Source analysis further mitigates this confound and allows reliable estimates of the data both at descriptive and inferential levels.

In sum, both factors (i.e., spatial filtering, head movements) lead to higher SNR in source-level as compared to sensor-level analyses (see also peak decoding accuracy in Figs. 5k vs. Supplementary Fig. 9b). We clarified this on pp. 45-46.

We are very sorry about a plotting error that occurred in Fig. 5i that shows the grand average (mean \pm SEM across subjects) of the classification metric with respect to time. We have erroneously included the SEM of the last participant instead of the SEM of the mean decoding accuracy across all 28 participants. This led to a distortion of the entire curve. This is now corrected. Decoding still peaks after cue onset but it is clearly above chance before cue onset.

However, it is correct that curves for trial-based decoding accuracy across time look different for rat multiple single-unit and human MEG data. Whereas decoding accuracy based on prefrontal single-units peaks at the time of choice, MEG-based decoding peaks in the second following cue onset. Findings from the literature indicate that these differences could be related to recording modality. As discussed above, it has been previously shown that rules can be decoded from both primate and rodent prefrontal cellular activity during all trial phases (Durstewitz D et al., *Neuron* 2010; Mansouri FA et al., *J Neurosci* 2006). Decoding of rule information is also possible from more macroscopic electrophysiological prefrontal signals like primate local field potentials (Buschman TJ et al., *Neuron* 2012) or human EEG (Collins J et al., *J Neurosci* 2014) but in this case significant decoding is only prominent following stimulus onset.

9. *As reported in their rodent data there is reference to decoding accuracy being above chance prior to cue onset. However, the data seem to show an increment in accuracy at cue onset – is this a significant increment. The figure related to the MEG data itself is not very informative (two small panels). The relevant figures are not well prepared – for example, the authors should indicate in the figure 5i what chance level (perhaps it is there but poorly reproduced).*

Please see the previous comment. Decoding is above chance prior to cue onset (i.e., the lower confidence interval of the curve is always higher than 33%), chance level is now indicated by a horizontal line. Sorry for this omission. However, it is correct that decoding accuracy increases following cue onset and this increment is indeed significant. We used a cluster-based permutation test comparing decoding accuracy following stimulus presentation to the pre-

stimulus baseline (Maris E et al., *J Neurosci Methods* 2007). Shaded areas in Fig. 5j correspond to time points for which the H0 hypothesis of no difference between decoding accuracy in baseline vs. task intervals was rejected.

10. It is not entirely clear how decoding accuracy was tested prior to implementation of this analysis? How was it tested? What precisely is a “task feature tested for relevance by a strategy”? I doubt this will be understood by the readership.

Thank you for pointing this out.

What is decoded? Attention acts like a filter such that only part of the state space (Fig. 2a, Fig. 6) is relevant for choice and learning when individuals test different hypothetical rules (e.g., *go-light*). This sub-space corresponds to the current attentional focus (e.g., the task feature cue light). See adapted Fig. 2a for a mapping of all strategies and corresponding state spaces. We decoded three different classes (attention either focused on visual, auditory or place cue) because they were detected in a majority of subjects (28/29) and thus a random-effects group-level analysis was possible. We agree that the term “task feature tested for relevance” is confusing and now instead use the term “current attentional focus” that we define in the Results section (p. 10) and now also provide a corresponding decoding analysis in rats (new Fig. 4e).

How was it tested? MEG data preprocessing involved the segmentation of epochs from 2s before to 4s after stimulus presentation. Eye-movement-related activity and cardiac signals were identified with independent component analysis and then discarded. The classification metric was then computed at the single-subject level. More specifically, classification was performed using linear discriminant analysis implementing ten-fold cross-validation with two repetitions. This means that the following process was performed twice: the entire data set was split into ten parts, with nine parts serving as a training set and one part as a test set. This was repeated ten times such that each part served as a test set once. Thus, the number of trials used for decoding depended on the total number of available trials per category. We computed discriminative information across time to test whether decoding accuracy depends on sensory stimuli and used a searchlight approach to highlight the source-level topography of decoding accuracy. We clarified this in the Methods section (pp. 46-47).

Reviewer #2 (Remarks to the Author):

Summary

This paper provides an elegant comparison of rule learning strategies across rats and humans. The authors show that rather than inefficient state-to-action learning in high-dimensional environments, learning performance in rats and humans indicates that both species reduce the dimensionality of the task space, thereby learning a low dimensional representation that allows for efficient decision-making despite changes to task rules. In addition, the authors show that by modelling attention during choice, they can explain sudden transitions in behavior to select and pursue a particular behavioral strategy. The authors implement cross-validated model comparison to demonstrate that attention at choice provides the winning model for behavior on their task. In addition, the authors show that behavioral strategies can be decoded from PFC in both species, using single-unit recordings in rats, and MEG data in humans.

Overview

This type of cross-species work is rare and challenging. I commend the authors for acquiring data in both humans and rats, using a comparable and non-trivial task. While previous studies have shown evidence for abstract rules being represented in PFC, here the authors provide insight into how attention may play a critical role in guiding behavior that draws from a low-dimensional representation. The behavioral analysis is careful and informative. The neural analysis could potentially be elaborated to strengthen the paper. Below are a few comments and suggestions to consider.

Thank you for your positive and supportive feedback. In our revision, we added more electrophysiological analyses that show how strategy decoding compares to other task parameters, link behavioral modeling results to prefrontal representation in rats, and strengthen the link between rat and human electrophysiological results (new material added to Figs. 4, 5 and new Supplementary Figures 8, 9).

Comments

1. Training (a clarification point): It is not clear how much training the rats received – when you say ‘rats were not pre-trained on any rule’, does this mean that they received no training at all, or they received training on other rules, or different orders of rules? Please can the authors make this clearer, in the Results section in particular. Related to this point, how do the authors account for an order effect? Is the order of task rules counterbalanced across subjects, or kept consistent?

Thanks for pointing out where our text needs further clarification. Rats received lever press training and they were exposed to the cues but they had no prior training with respect to any rule (see Methods pp. 34-35). The order of task rules was the same for all rats performing multiple rules (see cartoon added to Figure 2a). We added this information to the Results section (p. 5).

2. Cognitive process (a clarification point): if subjects are not pretrained, can the authors be clearer still on how the representation of the eight low-dimensional behavioral strategies are learned? The authors state that ‘many rats did not use the correct strategy before it was required’, and ‘rats learn new strategies if the behavioral repertoire does not match current task demands’. Does the behavioral modelling therefore show that different strategies are acquired according to the experimenter-defined order of task rules, such that some are represented early on, while others are only represented later on in the task? If this is the case, then can we really consider the rats to be performing hypothesis testing across all eight different behavioral strategies, if they only acquire an eight dimensional representation at the end of the experiment?

We do not assume that rats learn by performing hypothesis testing across a fixed set of behavioral strategies. This is assumed in serial hypothesis testing models (e.g., Wilson RC & Niv Y, *Front Hum Neurosci* 2012; Song M et al., *PLoS Comput Biol* 2022). However, our behavioral results (was Fig. 3 h-j, now Supplementary Fig. 7b-d) indicate that at least some strategies have to be learned by most rats (like *go-silent*).

Instead, our working hypothesis based on our data is that there are two levels at which the task is performed (see also new summary Fig. 6):

1) a lower level at which state-action values are learned within the attended-to sub-state space and strategies map to distinct state spaces via their attention focus.

2) a higher level at which strategies are selected from a pre-existing repertoire or created anew.

We only model the lower level using attention-modulated RL (Fig. 2, 5), but provide evidence for the second level through behavioral (Fig. 3) and neural decoding analyses (Figs. 4, 5). It has been proposed that the current hypothetical rule that is evaluated at the higher level is the source of top-down attention (Radulescu A et al., *TICS* 2019), but we are not aware of any model in the literature that combines features of an attention-based and a hierarchical RL model.

We made text revisions (Discussion pp. 27-31) to make this important and complex point clearer.

3. What neural computation is being modeled here? To perform this task, subjects need a representation of the low-dimensional state space, and a means to sequentially test and evaluate different options, before selecting and opting for a particular strategy. In other words, there are multiple processes at play here. Can the authors more clearly articulate which neural processes they are trying to account for in PFC? Are we observing changes in sampling of an established state space, or is the state space itself represented differently as the task proceeds?

This is an important point. As mentioned in the comment above, we assume that there are two different levels at which the task is performed. We provide evidence for two PFC representations using neural decoding analysis.

At the higher level, strategies can be considered *hypothetical rules* that summarize different combinations of environmental cues, actions, and potential outcomes within a common concept. In hierarchical RL model terms, strategies can be considered abstract actions with an established state space (see Fig. 6; Collins AGE & Frank MJ, *Psychol Rev* 2013). We have shown this in rat prelimbic recordings and now added a supplementary analysis to show that strategies can also be decoded from human MEG recordings (new Supplementary Fig. 9c, d).

At the lower level, attention reduces dimensionality because it acts like a filter such that only part of the state space is relevant for choice and learning (Fig. 2a). This sub-space corresponds to the *current attentional focus* and can also be decoded from a prefronto-parietal network in humans (Fig. 5k). Again, we added a corresponding analysis in rats (new Fig. 4e).

An additional neural computation that was reported in Fig. 4g (now Supplementary Fig. 8) is addressed in the following comment.

Mouse data:

4. Hypothesis testing: For Fig 4d and 4g: the significant decoding of behavioral strategy independent of current action is reported during cue and following lever onset. What does this mean in terms of neural computation?

4d (now Fig. 4g,h): The analysis shown in Figure 4d was to further test the idea that the animal's choices are based on behavioral strategies (entire task sets) rather than single situation-action pairs, i.e. that decision making is based on the former type of representation rather than the latter. If the PFC were to encode simple motor actions instead of strategies we would expect to

find a significant decrease in decoding accuracy for the “different” condition (i.e., when the classifier used to decode two different strategies is trained on trials with left lever presses and tested on trials with right lever presses or vice versa). Instead, we found that the side of the lever is not relevant in the 3s prior and following cue onset. The side of the lever was only relevant for strategy decoding when the action is performed (significant difference between “same” and “different” for the 3s period following lever onset). We adjusted the respective section in the Results (p. 21).

4g (now Supplementary Fig. 8): The analyses were performed to test whether rats have an abstract representation of the rule-relevant part of the state space even before performance increases can be detected. Formally, we show that the side of the lever press can be decoded during the task phases following cue and lever presentation if the cues that are relevant for the task rule are on the same side in training and test data sets. This can be considered as evidence for an abstract neural representation of the state space that is relevant for the task rule as predicted by models of structure learning (Tervo DG et al., *Curr Opin Neurobiol* 2016). After careful re-evaluation, we decided to transfer this part to the new Supplementary Fig. 8 because it is not the main thread of the paper and adds unnecessary complexity. Instead, we show several additional analyses that elaborate and strengthen our main electrophysiological findings (new Fig. 4c-f).

5. Would the authors suggest that a brain region other than PFC is performing hypothesis testing prior to choice?

Our neural analyses indicate that the current hypothesis (i.e., to follow a specific strategy) can already be decoded from the PFC prior to cue onset (Fig. 4). This is in line with primate and rodent single-unit data on prefrontal rule representations in the steady-state and it has been proposed that this could be a neuronal correlate of short-term memory of the task rule (Durstewitz N et al., *Neuron* 2010; Mansouri FA et al., *J Neurosci* 2006). From this perspective, it is expected that the same prefrontal strategy representations already exist prior to cue onset.

However, there are of course additional computational processes that can be considered aspects of hypothesis testing like the decision to stay with or switch away from the current strategy which we do not address in our study (see e.g., Koechlin E, *Philos Trans R Soc B: Biol Sci* 2014 for an overview of proposed computational processes). These additional computational layers involve multiple different (prefrontal) regions and are of course much more complex in humans.

We acknowledge these complexities in the Discussion section pp. 27-31 and the new summary Fig. 6.

6. Differences in PFC subregion: Were there differences in the neural code between Cg1, PrL, IL? What is the average response of these regions to the task? As a sanity check, it would be reassuring to see some more standard physiological analyses for ephys data acquired from these regions.

We apologize that the figure was not clear enough. Rat recordings were restricted to the PrL based on previous literature on encoding of rules in the steady-state (e.g., Durstewitz D et al., *Neuron* 2010). We adjusted Fig. 4a to make this clearer (PrL is now highlighted).

We agree that a better connection to the existing prefrontal literature is helpful. We therefore added the following new analyses. At the single-unit level, we show the PSTH of a unit with firing rates for two different strategies of the example session in Fig. 4b and a histogram with selectivity index d' -values for all units of that session (new Fig. 4c). At the population-level (new Fig. 4d-f), we systematically compared how decoding strength for strategies compare to that of cues (e.g., right vs. left cue light) and simple actions (i.e., side of lever press) for all trial phases. We also added analyses to make a stronger connection to our attention-modulated RL models (strategy decoding depends on attention-at-choice values, comparison of strategy decoding with (cue,act) pair decoding) and to the human MEG findings (decoding of the current attention focus as in Fig. 5i-k).

Human data:

7. Classifier: Is it not possible to train and test on the strategies that are included for a given subject? The decoding of cue prior to onset shows predictive activity but doesn't necessarily show evidence for coding of strategies.

While subjects on average tested most of the behavioral strategies (6 (5-6) out of 8 strategies), the specific types of strategies were not identical across subjects. A random effects model was therefore not possible. However, we now added a source-level fixed effects analysis (new Supplementary Fig. 9c, d) to address this point. More specifically, given that the data was expressed within the same geometrical position (i.e., parcels) within the same coordinate system, we pulled all trials of all subjects first. This resulted in a data matrix consisting of 11179 samples, 68 parcels, 976 time points, and 8 classes. This procedure resulted in the following distribution of class frequencies: 12.99% [go-light], 17.48% [go-dark], 17.45% [go-click], 21.50% [go-silent], 12.00% [go-right], 1.57% [go-left], 14.41% [alternate], 2.61% [win-stay lose-shift]. Supplementary Fig. 9c shows the decoding across time and Supplementary Fig. 9d shows the source-level topography of the highlighted decoding peak (averaged over selected time window: 180-280 ms after cue onset). The decoding peak was located in the right cingulate (caudal anterior division and posterior cingulate). In the context of rule learning and switching, anterior cingulate seems to be involved in monitoring the reliability of the current behavioral strategy (Koechlin E, *Philos Trans R Soc B: Biol Sci* 2014). Significant decoding was also detected in left superior frontal gyrus, right precentral gyrus, right postcentral gyrus, left inferior parietal cortex and left lateral occipital cortex.

8. Significance of the effect: Fig 5h: The distribution of the data across subjects is not shown. Please show the distribution of individual data points and make clear how significance is determined.

Thank you for raising this issue. The panel you refer to (now Fig. 5k) illustrates the spatial topography of the decoding results averaged over 0 to 1 s post cue onset (which is the decoding peak across time) at the group level. The classification metric was computed at the single-subject level across time and the distribution of the data across subjects is shown in the corrected Fig. 5i as the grand average (mean \pm SEM) of the decoding accuracy across time. We now also show an overlay of all 28 individual decoding curves in Supplementary Fig. 9a.

Decoding is above chance already prior to cue onset (i.e., the lower confidence interval of the curve is always higher than the 33% chance level). Moreover, we added a cluster-based permutation test (Maris E et al., *J Neurosci Methods* 2007) to compare decoding accuracy

following cue onset to the pre-stimulus baseline. Indeed, decoding accuracy significantly increased following stimulus onset. Shaded areas in Fig. 5j correspond to time points for which the H0 hypothesis of no difference between decoding accuracy in baseline vs. task intervals was rejected.

9. Given that large areas of the brain appear to show significance it isn't clear that this is an effect specific to PFC.

We agree that the discriminative activity shown in Fig. 5k (previously Fig. 5h) is not PFC-specific but the topography is reminiscent of the fronto-parietal control network that has also been implicated in rule-based attention switches before (Niv Y et al., *J Neurosci* 2015).

We address this point in the Discussion (p. 31).

10. Moreover, it isn't clear which time bin this data comes from and how the time bin was selected (with multiple comparisons in mind).

The topography is based on the averaged time window of 0 to 1 second after stimulus onset that includes the peak in discriminative activity as shown in Fig. 5i. As detailed in response to comment 8, we also added a cluster-based permutation test (Fig. 5j) to show that in this time window, decoding accuracy is indeed significantly higher. We made this point more salient in the figure legend and Results section (p. 25).

11. Is this prior to cue onset as suggested to be the case in rats? Similarly, Fig 5i: how is significance assessed here? Please indicate the chance level on the plot. Is this using decoding of relevant features, strategies, or something else?

We decoded the *current attentional focus*, i.e., the attended to part of the state space that is relevant for choice and learning (see Fig. 2a for a definition). We also added a horizontal line at 33.3% chance level (now Fig. 5i) to indicate when significance is reached. Sorry for this omission. Decoding accuracy is clearly above chance prior to cue onset as in rats. Please see comment 8 for further details on how significance was assessed. We clarified this in the Methods (pp. 46-47), figure caption and legend.

Cross-species comparisons:

12. Task design: From Fig 5a it is hard to understand what the human task involves and how it provides a direct comparison to the one used in rats. Can the authors provide a more detailed schematic and highlight points of comparison across species?

Thank you for this comment. We added a schematic to Fig. 5a and summarize points of comparison across species in the revised Discussion (pp. 27-31, summarized in Fig. 6).

13. Learning curves: While the authors show the learning curves for models, they do not show

the equivalent learning curve for raw behavioral data from rats (except for Fig 1d) and humans. Please can the authors add this in to help provide qualitative comparison of the step changes in subject learning vs ACL model learning?

This is an important point. We added color-coded learning sigmoidal learning curves based on performance that indicate that there are transitions instead of gradual learning (Figs. 1d, 2e-g, 5f-h).

14. Analyses: Can the authors provide more direct comparisons across species? For example, does the reaction time effect in humans replicate in rodents? Does the multiple regression analysis applied to rodent behavior pull out the same results in humans?

Thank you for bringing up this issue. Please see the detailed response to the first comment from reviewer #1 who raised a very similar issue. There, we also describe several additional analyses we conducted to strengthen translational links.

However, reaction time effects found in humans cannot be observed in rats. This is due to the experimental setting: rats are usually constantly moving in the operant chamber and therefore have a different distance and orientation with respect to the lever they choose in each trial. Variance in reaction time due to locomotor effects is thus very high and obscures potential “cognitive” effects on reaction time.

Unfortunately, behavioral time series in humans are much shorter and therefore a corresponding multiple regression analysis with the same parameters is not possible. This is a limitation but was necessary to be able to fit multiple rule switches into one MEG session.

We summarize points of comparison across species in the revised Discussion (pp. 27-31).

Reviewer #3 (Remarks to the Author):

Species-conserved mechanisms of abstract rule learning promote cognitive flexibility in complex environments

Bähner, Popov, Boehme, Hermann, Merten, Zingone, Koppe, Meyer-Lindenberg, Toutounji, and Durstewitz

This manuscript investigates rapid learning in complex environments, a fundamental aspect of intelligent behavior. Humans are thought to use abstract strategies to focus on particular aspects of a task to speed learning (sometimes referred to as ‘hypothesis testing’). However, it remains unknown if similar mechanisms exist in other species. To address this question, the authors used a combination of behavioral, computational, and electrophysiological analyses in both rats and humans to study how subjects learned new tasks. Rats performed a multidimensional task with four possible features guiding response: auditory stimulus, visual stimulus, reward history, and choice history. During each block of trials, subjects had to

discover the relevant feature dimension through trial and error. Humans performed a similar task variant. Using statistical techniques and computational modeling of behavior, the authors found evidence that both species sequentially sampled different ‘strategies’, with a strategy defined as what task feature was being attended. In other words, the subjects were not learning the Q-value in the full dimensional feature space. Furthermore, by combining electrophysiological measurements in rats and magnetoencephalography (MEG) in humans with classifier analysis, they demonstrated that the prelimbic cortex in rats and the prefrontal cortex in humans encode abstract task strategies.

Overall, this is an interesting study with clever task design and rich dataset. The manuscript is clearly written with comprehensive review of relevant literature in the introduction. The results are exciting and consistent with a growing body of work that suggests learning does not necessarily occur at the level of raw features but can take advantage of lower-dimensional representations.

Thank you very much for your positive and encouraging feedback.

However, despite our general enthusiasm, we do have some comments:

Major comments:

1a. The authors' find rats used strategies even when the reward feedback was random. If true, this provides strong evidence that rats sample different strategies but also needs stronger verification. It would be helpful to include a figure depicting the behavioral performance of naïve and non-naïve rats with random reward feedback, along with an analysis demonstrating that these animals attend to stimuli if their detected strategy includes a stimulus.

Thank you for bringing up this point. Several points can be made to address this issue:

First, we added a new analysis to show that more strategies can be detected in the naïve random reward cohort as compared to strategies detected in synthetic data with the same number of trials based on a random agent (random selection of lever side and random reward feedback with equal probability). This is shown in Fig. 1c, explained in the Methods section p. 40 and addressed in detail in one of the following comments.

Moreover, we had already shown that head movements in naïve rats with random reward predict choice 3s later (see Supplementary Fig. 3). This is a strong indication that behavior is not random but goal-directed. We added a corresponding figure with head direction plots for experienced rats receiving random reward with comparable results (new Supplementary Fig. 4). Furthermore, we compared the median attention-at-choice values in both random reward cohorts and found no significant difference ($p=0.37$, Mann-Whitney test, new Supplementary Fig. 7a).

Finally, results in Fig. 3e already highlighted differences in the type of strategies selected in naïve vs. experienced rats receiving random reward feedback and these strongly depend on prior learning experience. We now provide an additional analysis (Supplementary Fig. 7a) where we compare the frequency of each strategy in naïve vs. experienced rats performing the random rule.

1b. It also isn't clear why naïve rats would employ any strategies. If the animal does not understand the levers lead to reward, then why would they push the levers at all? Were they also pretrained to push levers for reward?

Yes, all rats were trained to press levers for reward before the rule learning experiment started (described in Methods pp. 34-35, now highlighted in the Results section, p. 5). There is also evidence in the literature (Evenden JL & Robbins TW, *The Quarterly Journal of Experimental Psychology B* 1984; Dember WN & Fowler H, *Psychol Bull* 1958; Maggi S et al., *Elife* 2024) that rodents spontaneously use some strategies like *win-stay-lose-shift*, *place*, *alternate* or that they test the relevance of salient stimuli (Floresco SB et al., *Behav Brain Res* 2008). It has been proposed that these are exploratory behaviors with high ecological validity (See Introduction p. 4 and revised Discussion p. 28). For humans it has also been reported that hypothetical rules are explored even if there is no direct behavioral advantage (Collins AG et al., *J Neurosci* 2014; Maggi S et al., *Elife* 2024).

1c. If so, then one might expect they engage in the random response, perseveration, and Win-Stay Lose-Shift (WSLS) strategies but it still isn't clear as to why they would respond to stimuli that were meaningless (even if they were familiar with the cues).

In the introduction of the manuscript, we hypothesize based on previous literature (Evenden JL & Robbins TW, *The Quarterly Journal of Experimental Psychology B* 1984; Dember WN & Fowler H, *Psychol Bull* 1958; Maggi S et al., *Elife* 2024) that strategies are an exploratory behavior and it may be a useful and efficient way to sample environment for relevance. This hypothesis was motivated by the fact that some strategies have repeatedly been reported in both rats and humans in the literature and we therefore speculated that they might be useful heuristics that are used more broadly during rule learning. It has also been suggested that such an approach could facilitate generalization of task rules in humans (Collins AG et al., *J Neurosci* 2014).

Head direction plots indicate that choices are not random but goal-directed even if reward feedback is random (Supplementary Figs. 3, 4). We also added an analysis where we show that rats and humans receiving randomized reward feedback do not behave like a random agent (see Figs. 1c, 5b).

We do find that perseveration occurs at the strategy level (in the sense of a negative transfer effect) which negatively affects rule learning (see Fig. 1e, Supplementary Table 2 and Fig. 3c).

Surprisingly, WSLS was not a preferred strategy in either rats (Supplementary Fig. 7a) nor in humans (see Figure legend of Supplementary Fig. 9b,c).

We added a paragraph to the discussion that addresses this issue (p. 28).

1d. Furthermore, it isn't clear as to why the animals would have this basis set of strategies. These strategies don't seem particularly ecologically relevant given the artificial nature of the stimuli.

Four of the strategies (*place* strategies, *alternate* and *WSLS*) can be observed in multiple experimental settings which indicate that they are ecologically relevant (see also previous comments). Rats are also curious to learn about the relevance of task stimuli in an operant

chamber but it is more difficult for them to learn an anti-cue rule (Floresco SB et al., *Behav Brain Res* 2008). The natural tendency to follow these strategies was also observed in our data from naïve rats in the random rule (see Supplementary Figure 7a).

2a. Related to the above comment, the ability to identify strategies should be more completely verified. For example, were the same strategies found in simulated data from a randomly responding model?

Thank you for pointing this out. We therefore directly compared behavior in the random rule against strategies detected in simulated data with randomized choices (10,000 sessions with the same length as in rats and humans, see Methods p. 40). In both rats and humans, the percentage of trials that can be explained by strategies was significantly higher for naive subjects performing the random rule task (see Figs. 1c, 5b).

2b. Or simulations from a model that was using a few of the strategies (e.g., just win-stay-lose-shift or just a subset of the strategies). Or in the data from the model that was designed to sample from different strategies?

Some quantification of the ability of the algorithm to reliably identify sequences of strategies would be helpful.

Thank you for raising this issue. In order to quantify the rate of false-negatives, we used the following approach: we created sessions that consisted of 100 trials with random choices. Next, we inserted one synthetic strategy sequence per session at randomized positions and counted the percentage of strategy trials that was correctly detected by the strategy detection algorithm (i.e., the sensitivity of the detection algorithm). This process was repeated 10,000 times (1250 times per each of the eight strategies) for multiple sequence lengths (ranging from 6-50 trials). The results are presented in the new Figure 1c and show that a sensitivity of >80% is reached for sequence lengths of ≥ 9 trials.

3a. The authors assert that their defined strategies are lower-dimensional than the full space of Q-values. Isn't this only true because the authors selected a subset of strategies to test? Other strategies, such as arbitrary associations of light position, sound, and previous response, would greatly expand the space of strategies. How were these strategies chosen?

Even in a confined experimental setting like in our case, one may think of endless combinations of stimuli and actions other than those tested that could be used as a strategy. Our selection was based on previous literature (see comments above) and the hypothesis that rats use those strategies to test task features for relevance. This would help to explain experimental effects such as sudden transitions in behavior as well as transfer effects that have been experimentally observed both in rats and humans (See Introduction). Our results and additional analyses discussed in comment 2a provide evidence that rats and humans actually use these strategies. However, we agree that other, high-dimensional strategies could also be used to learn task rules. We address this concern below (see comment 3d).

3b. Were all necessary to explain the data?

Yes - we selected the task rules such that we could check whether rats use the proposed strategies to test task features for relevance (e.g., see Figure 1e, f). See Figs. 1b, 2a for a mapping of strategies to task features.

3c. Did expanding the set of models improve behavioral fits?

Yes, this was indeed the case. We found that both in rats and humans, the ACL model was the best at predicting held-out behavioral responses (Figs. 2d, 3f, g, 5e). Moreover, only models that had an attention-at-choice component were able to reproduce the experimental observation of sudden transitions (Fig. 2c).

To address these concerns, we would suggest the authors:

3d. Incorporate mixed strategies into their strategy detection algorithm and demonstrate that rats do not use these more complex strategies but prefer the low-dimensional eight strategies defined in the task.

Thank you for this important comment. The most straightforward way to implement mixed strategies would be to use a *win stay-lose shift* vs. *win shift-lose stay* approach with respect to place, visual and auditory cues. It is possible to learn all task rules with these six strategies and sudden transitions in behavior would also be conceivable. However, these alternative strategies are high-dimensional compared to our proposed set of eight strategies because they all depend on two task features (e.g., win stay-lose shift visual depends on the current light cue and the previous reward).

In order to test this alternative hypothesis, we adapted the strategy detection algorithm to directly compare rat and human behavior in the random rule against strategies detected in simulated data with randomized choices (10,000 sessions with the same length as in rats and humans, see Methods p. 40). In both rats and humans, the percentage of trials that can be explained by alternative strategies was not statistically different from the random agent (Figs. 1c, 5b). Moreover, a higher percentage of trials could be assigned to the proposed, low-dimensional vs. the alternative, high-dimensional set of strategies in both rats and humans performing the random rule task (Figs. 1c, 5b).

Interestingly, the one strategy that was part of both sets of strategies (*win stay-lose shift* with respect to place) was not a preferred strategy in either rats (Supplementary Fig. 7a) or humans (Supplementary Fig. 9b,c).

3e. Demonstrate that the reduction in dimensionality is due to attentional mechanisms. Specifically, when rats employ a specific strategy, trial-by-trial behavioral choices should correlate with the attentional factors defined in their model (WAC and WAL). Conversely, trial-by-trial behavioral choices should weakly correlate with attentional factors when animals use mixed strategies.

Even though there is little evidence for mixed strategies (see above), several points can be made to address this issue.

Head direction plots highlight that attention is focused on a specific task feature in a strategy-specific manner (Fig. 2, Supplementary Figs. 2-4) and attention-modulated RL models indicate that they serve to reduce dimensionality (because ACL is the winner model). In order to make this point more compelling, we used computational modeling to show that rats are in a “tunnel state” with a very strong focus on only one task feature (see next point for results of this additional analysis).

Moreover, additional results on positive transfer effects (see new Fig. 3d) indicate that a stronger attention focus is a predictor of learning speed (i.e., stronger dimensionality reduction is adaptive).

We made this important point more salient in the Results section (pp. 10-12, 15-16).

4. It wasn't clear to us how the ACL model, where attention acted on both the choice and learning, differed from attention filtering of the stimulus input. This seems like the more common way of thinking about attention – that it filters sensory stimuli, allowing both learning and choice to be driven by the attended feature. If these models are not equivalent, then we would suggest fitting a new model where attention acts on sensory inputs to both choice and learning.

The binary ACL model as used in humans would be an exact equivalent of a filter. The ACL model used in rats (based on empirical attention values) is more like a graded filter. However, both model variants give similar results and the filter-like effect is responsible for sudden transitions. We now included binary models into the set of RL models tested on rat data. Both ACL and binary ACL were the winning models and there was no statistical difference between the two ($p=0.87$, Figs. 2d). This finding shows that attention is indeed a very strong filter and thus evidence for low-dimensional strategies in both species.

5. Related to the previous comment, were there any differences in the attentional modulation between species?

In both species, ACL was the winning model and the attention-at-choice component was necessary for step changes. However, since no eye tracking data were available for the human cohort, we cannot investigate more subtle differences. This should be addressed in future studies. We make this explicit in the Discussion (p. 31).

6. *If rat prelimbic cortex represents the animal's abstract strategy, then was the strength of the strategy classifier (Fig. 4c) correlated with the attentional factors estimated by the model?*

Thank you for this important comment. In order to address this point, we tested whether the median attention-at-choice for a pair of strategies was correlated with the decoding accuracy in the task stage following cue onset. There was a significant correlation ($p=0.006$) after correcting for the number of neurons recorded in this session ($p=1.7*10^{-5}$, $R^2_{\text{adjusted}}=18.2\%$). The correlation between attention-at-reward and decoding accuracy in the corresponding task stage was not significant ($p=0.09$) after correcting for the number of neurons recorded ($p=2*10^{-4}$, $R^2_{\text{adjusted}}=11.8\%$). We now present these results in Fig. 4d.

7. *Based on Figure 1d, it seems that rats will occasionally adopt the correct strategy before abandoning it. Why would that occur? In the authors' model, once the correct strategy is identified the animal should persist, which is what leads to the rapid learning observed in the animal's behavior.*

This is correct and relevant for learning as indicated by our regression model shown in Fig. 1f. The literature on hierarchical RL (Collins AG & Frank MJ, *Psychol Rev* 2013) assumes that strategy selection is value-driven. However, the exploration-exploitation balance is also relevant for action selection and our data indicate that rats are not strictly exploitative (e.g., see detected strategies in task rule alternate in Fig. 2e). We intend to model the full hierarchy in future work where we will also account for the balance between exploration and exploitation at the level of strategies, not only actions. To the best of our knowledge, no RL models currently exist that combine features of an attention-based and a hierarchical RL model. We addressed this issue in the Discussion (p. 30).

8. *The authors use the animal's body orientation to help identify the strategy they were using. However, this raises the concern that neurons in prelimbic cortex are not encoding strategy per-se but rather the body position, etc. that is being driven by the strategy. For example, previous work has shown that neurons in the rat prelimbic cortex encode spatial locations (Hok et al., 2005, PNAS) and the author's show that the animal's head direction changed based on the strategy (Fig. 2a,b). Is there any way to exclude these potential confounds in Figure 4? Similar questions could be asked for human participants. If not, then perhaps the authors can discuss these concerns in the Discussion section.*

This is an important issue and several points can be made to address this issue.

We don't use body orientation to identify the strategy, only the choice behavior and our detection algorithm. We find, however, that body orientation is an indicator of the strategy.

Moreover, the analysis now shown in Figure 4g, h (was 4c, d) more directly addresses this issue. If the rat PFC encodes simple motor actions instead of strategies we would expect to find a significant decrease in decoding accuracy for the "different" condition (i.e., when the classifier used to decode two different strategies is trained on trials with left lever presses and tested on trials with right lever presses or vice versa). Instead, we find that the side of the lever is not relevant in the 3s prior and following cue onset. The side of the lever is only relevant for strategy decoding when the action is performed (significant difference between "same" and "different" for the 3s period following lever onset).

In humans, this confound is not relevant because motor responses are stereotypical.

We now also address this point in the Discussion section (p. 30).

9. Were classifiers trained on balanced number of trials of each experimental condition? In general, we would suggest including more detail in the description of the classifier in the Methods section.

Thank you for this comment. We added additional information to the Methods section (see pp. 44, 46). Below, we summarize the basic analysis pipeline.

Decoding of corresponding behavioral information from rat multiple-single unit and human MEG signals was performed but the analysis parameters were not identical. Selection of analysis parameters was based on differences in the signal type, data structure (usually multiple sessions per rule in rats vs. only one experimental session comprising five different rules available in humans), analysis goals (session-wise population decoding in rats vs. random-effects group analysis in humans) and on recommendations given in the respective toolboxes (multiple single-unit data: Neural Decoding Toolbox; decoding of MEG signals: MVPA-Light toolbox).

In rats, analyses were run using neural data from individual experimental sessions (using recordings from both the multidimensional rule-learning task and the probabilistic two-rule set shift). We included all sessions with ≥ 10 simultaneously recorded neurons and at least two different strategy types with ≥ 15 trials each: 62 experimental sessions from nine rats. Raw spike trains were first aligned to trial onset for each prefrontal neuron. Trial-aligned spike trains from trials of a behavioral category were concatenated for each neuron and labelled accordingly. Spikes were then binned using a sliding window approach (bin width 3000ms, step size 300ms). We used z-score normalized data to avoid that neurons with higher firing rates have a larger influence on the classification results. We used a maximum-correlation-coefficient classifier for decoding analyses based on leave-one-out cross-validation. The classifier learns a mean neural population vector for each class (i.e., a template for a behavioral category) by averaging all training points within each class. The classifier predicts the correct class using Pearson's correlation coefficient between the test data and the training set of each class (the highest correlation value corresponds to the predicted label). At least 15 trials per each behavioral category were required: neural data from 14 trials were used as a training set to predict behavior in the test trial. Decoding accuracy for each session was based on 50 cross-validation runs using a resampling procedure (i.e., 15 trials per category were randomly drawn from the pool of all available trials per behavioral category in each run).

In humans, decoding analyses were performed at the single-subject level based on data from 28 subjects. MEG data preprocessing involved the segmentation of epochs from 2s before to 4s after stimulus presentation. Eye-movement-related activity and cardiac signals were identified with independent component analysis and then discarded. Classification was performed using linear discriminant analysis implementing ten-fold cross-validation with two repetitions. This means that the following process was performed twice: the entire data set was split into ten parts, with nine parts serving as a training set and one part as a test set. This was repeated ten times such that each part served as a test set once. Thus, the number of trials used for decoding depended on the total number of available trials per category and was not fixed as in rats. We computed discriminative information across time to test whether decoding accuracy depends on sensory stimuli and used a searchlight approach to highlight the source-level topography of decoding accuracy.

10a. *The authors repeatedly claim that rats learned rapidly. However, they experience hundreds to thousands of trials before learning (e.g., Figure 1d). Why should we consider this to be rapid learning? How should we interpret this relative to the dozens of trials that humans require?*

We agree that our wording is misleading. Learning in rats is of course slow as compared to humans but *transitions in behavior* are rapid as indicated by sudden changes in performance shown in change point analyses (see main text p. 5, Figs. 1d, 2e-g), and as has been observed in several studies before (Durstewitz et al., *Neuron* 2010). We made this point clearer in the Discussion (p. 28, see also legend of Fig. 6).

10b. *Related to this comment, it isn't clear as to why performance in naïve rats declines over the initial four days (Fig. 1d)? Even if sampling strategies randomly, shouldn't it be stable until they hit upon the right one?*

We are sorry for this confusion. In Fig. 1d, the plot does not show current performance but performance changes. We used a so-called CUSUM (cumulative sum of differences to the mean; Basseville *M Automatica* 1988) curve to visualize the global minimum that indicates that performance increases with respect to mean performance after the change point. We agree that this type of visualization can be misleading and instead now fit a sigmoidal model to show that performance changes are sudden and not gradual during learning.

11. *How many strategy switches did the animal's experience before they stabilized on the correct strategy? How many times did they visit each strategy? Did they revisit poorly performing strategies?*

These are important points that we partially address in the Supplementary Table 1 where we compare strategy-related quantities across rules. We made this point more salient in the Results (p. 7) because these questions were the motivation for the regression analysis shown in Fig. 1f that shows that strategy use predicts learning. Revisiting poorly performing strategies is a negative predictor of learning (Figs. 1f, 3f, Supplementary Table 2).

We added the number of strategy sequences rats experience before they stabilize on the correct strategy in different rules to the Supplementary Table 1. In order to address the second point, we provide an additional analysis (Supplementary Fig. 7a) where we compare the frequency of each strategy in naïve vs. experienced rats performing the random rule. We decided to use these cohorts because selection frequencies depend on reinforcement and learning history.

12. *On any given trial, there may be multiple feature changes that could lead to a change in orientation (such as seen in Figure 2a). Can the effect of different features be dissociated?*

Head direction plots are an average of hundreds of strategy sequences. We argue that averaging does control for potential contributions of other features to a change in orientation reaction because the number of features that change is varied systematically. However, for three strategies (*go-right*, *go-left* and *alternate*) we also found strategy-specific movement patterns even before cue onset similar to head direction plots, which indicates that rats focus their attention on a single strategy. More specifically, we found that rats tend to stay in the respective side of the operant chamber at the onset of a *place* strategy and move away directly after a

sequence stops. Similarly, rats switch sides in the inter-trial interval while they follow the *alternate* strategy. We detected this effect in three different data sets (Supplementary Fig. 5). This indicates a strong top-down process such that the attention selection process is already completed before cue onset.

13. It isn't clear how well the models fit the behavioral data. Figure 2d shows the ACL model fits better than others, but the overall quality of fit isn't clear. It would be useful to include the animal's behavior in Figure 2e-g for comparison.

We added color-coded sigmoidal learning curves based on performance data.

14. In Figure 4d, 4f, and 4g – what is chance level? We assume it's 50%?

Yes, this is correct for Figure 4d, f (now Figs. 4h, Supplementary Fig. 8b). We added a dotted horizontal line at 50% chance level. Fig. 4g (now Supplementary Fig. 8c) shows the difference in decoding accuracy between two conditions and the plot therefore has no chance level. We adapted the label of the y-axis to make this clear.

15. If so, then are all of these significantly greater than chance?

Yes, we added the following statistics to the respective figure legends:

- Fig. 4h (was Fig. 4d): Decoding accuracy was above chance in all conditions (all $p < 10^{-5}$, unpaired t-test).
- Supplementary Fig. 8b (was Fig. 4f): Decoding accuracy was above chance in all conditions (all $p < 10^{-4}$, unpaired t-test) with the exception of the last condition (i.e., data at the time of lever press with different task feature in training and test; $p = 0.95$).

16. While the differences between same and different are interesting, they can only be interpreted if at least one of them is greater than chance.

We are sorry that this was not clear enough in the text. We corrected the figure caption accordingly.

Minor comments:

1. What are the region names in Fig. 4a?

We added a legend for the abbreviations used in Fig. 4a to the figure legend.

2. *What is the range of performance along the y-axis of Figure 1d? It is important to clarify how much of the performance varies across time.*

This is the performance change normalized to one. Given that the interpretation of CUSUM curves is not straightforward (see comments on learning curves above), we decided to visualize all learning curves with a sigmoidal learning model.

3. *From Figure 1d, it seems as if strategies overlapped in time. Is this true? From our understanding of the approach, only a single strategy should be enacted at a time?*

We are sorry. This is a visual artifact we failed to correct. All figures with a visualization of strategy sequences are now presented as in Figs. 2e-g.

4. *In Figure 2a, when comparing across switch point – were the stimuli and response matched across trials? If not, could the observed results reflect these other changes rather than just the internal change of strategy?*

No, they were not matched but due to our experimental design, stimulus location was systematically varied. Head direction plots in Fig. 2 and the corresponding Supplementary Figures 2-4 are usually based on hundreds of strategy sequences. We provide multiple lines of evidence that we can exclude relevant changes in head direction on switch trials that are not due to internal change of strategy (see also response to major comment #12 for details).

5. *It would be helpful to indicate which horizontal lines in Fig. 2e-g represent the task switches and which indicate changes in the animal's strategy.*

Thank you for pointing this out. We changed the figure captions accordingly.

6. *It would be helpful to clarify the meaning of "multi-rule rats" in the title of Fig. 2d.*

We added a cartoon to Fig. 2a that visualizes the sequence of rules rats learned in this experiment.

7. *The trial labels are missing on the x-axis of Figs. 2e-g.*

Trial labels were added.

8. *In the legend of Figure 2c, what is "filled colored circles" referring to in line 235?*

Detected strategy trials are displayed as color-coded filled circles. Since they are densely packed and the resolution is not high enough, they appear like a color band. We decided to use this style to avoid the visual artifacts addressed in minor comment #3. We made this clearer in the figure legend.

9. The "go-silent" is not italicized on line 270.

To make the distinction between task rule and strategies clearer, we decided to italicize behavioral strategies. Go-silent refers to a task rule here. To visually support this distinction, we decided to present task rules with a rectangular shape and strategies with an oval-shaped icon in all figures (e.g., Fig. 1b).

10. The y-axis of Figs. 3h and 3j are not labeled.

Sorry for this omission. We added labels.

11. No statistics are provided to support the statement on line 345.

Decoding accuracy was statistically above chance prior to cue onset: $p=2.7*10^{-36}$ (one-sample t-test). We added this information to the Results section (p. 19).

12. What regions in Fig.5h are significant? What timepoints in Fig. 5i?

In Fig. 5h (now Fig. 5k), the following regions are significant: bilateral superior frontal gyrus, right middle frontal gyrus, right precentral gyrus and left superior/inferior parietal gyrus. We added this information to the Results section (p. 25). All time points for which the lower bound of the confidence interval exceeds chance level contain significant decoding information. After correction of a plotting error (see reviewer 1, comment 8), it can be seen in Fig. 5i that decoding accuracy is above chance across all trial phases (even before stimulus presentation). We also added this information to the main text (p. 25) and figure legend.

13. Should "alternate" be within the parentheses of "win-stay-lose-shift" in the supplemental information, line 94?

We agree that *alternate* could both be conceptualized as *win shift-lose-stay* and as *delayed alternation*. However, we have behavioral evidence from video tracking that place is the attended task feature when rats use the *alternate* strategy (see major comment #12). Moreover, we used this alternative conceptualization in the new set of alternative strategies and show that they are not detected above chance in both rats and humans (see Figs. 1c, 5b).

14. In Fig. S1b, what was the criteria used to define an animal as a "bad-learner"?

We did not attempt to categorize rats as good vs. bad learners based on statistics. The intention was to pick two examples of rats that differ with respect to learning speed and use Supplementary Figure 1 to visualize strategy-related quantities that were identified as significant predictors of learning in the regression model shown in Figure 1F. Given that the predictors of the regression model are already discussed in Supplementary Table 2 and Supplementary Figure 1 is thus both redundant and complex, we decided to drop that figure in the revised manuscript.

Reviewer #3 (Remarks on code availability):

I didn't try to execute the code, but I looked through the readme files and everything seems to be clearly explained.

Reviewer #4 (Remarks to the Author):

This manuscript explores computational and neuronal underpinning of abstract rule learning in rats and human participants using behavioral, electrophysiological and MEG experiments.

Unfortunately, the data, the analyses, visualization and arguments are not sufficiently convincing to support the conclusions, at the level necessary for Nature Communications. There are several issues with the paper:

1. The analyses presented across the manuscript are not performed in sufficient depth, they are not clearly motivated and they are often not separating different hypotheses from each other. For example, the first experiment/figure concludes that “rats test low-dimensional behavioral strategies rather than learning high-dimensional mappings between environmental states and actions”. It is unclear what analyses clearly establishes that rats use low-dimensional behavioral strategies. What in the experimental design and analyses separate/diagnose the use of low-dimensional behavioral strategies from that of high-dimensional behavioral strategies? How does this section/analyses leads to "selective attention" section (Fig.2)?

Thank you for your comment. Below, we summarize our line of reasoning for the hypothesis that rats use low-dimensional strategies and also make the link to selective attention more salient. We used our strategy detection algorithm along with behavioral modeling to show that several characteristics of the observed response behavior such as the presence of sudden changes in performance and the observed learning variance (Fig. 1d-f) can best be explained by the use of behavioral strategies. RL models showed that rats do not learn the full state-action-space or variants thereof (Fig. 2d). Selective attention is the proposed mechanism that reduces dimensionality during strategy use as compared to learning high-dimensional mappings between environmental states and actions. This is formally shown using RL modelling (ACL is the winning model, see Fig. 2d).

We adapted the Results section to make this clearer (pp. 6-7) and also added several analyses to strengthen our claims. The focus of Fig. 1 is now on evidence for low-dimensional vs. high-dimensional strategies in rats. This is also reflected by an adjusted figure title (now: “*Rats test low-dimensional strategies to infer the task rule.*”).

- We validated results of the strategy detection algorithm to show that strategies are not false-positives (rats performing a task version with random reward feedback do not behave like randomly choosing agents) and can be detected with sufficient power (false-negative rate: we used synthetic strategy sequences of varying lengths to evaluate the percentage of trials that was correctly detected by the strategy detection algorithm). These results are shown in Fig. 1c and the details are given in Methods p. 40 (see also from reviewer 3, comment #2).

- To address potential alternative explanations for the observed response patterns as pointed out in your comment (low-dimensional vs. high-dimensional strategies), we conducted the following analysis. We tested whether rats use alternative, higher-dimensional strategies that are suitable for rule learning and compatible with performance steps (*win stay-lose shift* vs. *win shift-lose stay* approach with respect to place, visual and auditory cues) and found that the rate of detected strategies was not higher in rats performing the random rule task as compared to a random agent (see Fig. 1c).
- The link between behavioral strategies and selective attention is further strengthened by showing that the binary ACL model performs on par with the continuous ACL model based on head movements in rats and both ACL models were better than all other models (Figs. 2d, 5e, Supplementary Fig. 6). This indicates that attention is a very strong filter resulting in strong dimensionality reduction (Results pp. 11-13). At the neural level, decoding accuracy of behavioral strategies is correlated with attention-at-choice values and significantly higher than (cue,act) pair decoding as defined as in the RL models (Figs. 4d, f; see also next comment).

2. The presentation and analyses of electrophysiological experiments does not meet the norm standard in the electrophysiology field. Several studies have shown the complexity and challenges in analyzing PFC data during complex behavioral tasks. This study brushes over this, only showing a few rather indirect and abstract figures to describe the data and results.

We added several analyses in rats and humans to address this point.

Rat data (see new material added to Fig. 4, new Supplementary Fig. 8):

- At the single-unit level, we show the PSTH of a unit with average firing rates for two different strategies for the example session displayed in Fig. 4b. We also provide a histogram with selectivity index d' -values for all units of that session (new Fig. 4c). This illustrates the known phenomenon that decoding of prefrontal information is weaker at the single-unit vs. population level and that above-chance decoding of strategies before cue onset can also be found at the single-unit level (Results pp. 19-20).
- At the population-level, we evaluated how decoding strength for strategies compares to that of cues and specific actions for all three trial phases (3s pre-cue, post-cue and post-lever onset). We also compared this against decoding of the current attention focus similar to what is shown for human MEG in Fig. 5i-k. A two-way repeated-measures ANOVA had a significant interaction effect ($F(6,54)=32.65$, $p<0.001$, multivariate results) and we found significant simple main effects for all task stages (pre-cue: $F(3,57)=85.5$, post-cue: $F(3,57)=90.6$, post-lever onset: $F(3,57)=99.6$, all $p<0.001$). Bonferroni-corrected pairwise comparisons showed that decoding of strategies and attention focus is comparable and outperforms decoding of cues and simple actions with most prominent effects appearing prior to cue onset. Decoding of cues was weakest in all follow-up comparisons (all $p<0.001$) and at the time of lever press, decoding accuracy was not different between strategies, attention focus and simple motor actions (new Fig. 4e).
- At the population-level, we provide further links between strategies and selective attention (models). We show that decoding accuracy is significantly correlated with the median attention-at-choice of the strategy trials that were used for decoding (Fig. 4d). Moreover, we use another two-way repeated measures ANOVA (interaction effect not significant $F(6,33)=2.3$, $p=0.056$, multivariate results) to test how decoding of strategies

across consecutive trial phases compares to decoding of state-action-pairs as defined in the 2D-RL, 3D-RL and 4D-RL models shown in Figure 2d. There was a main effect of the factors model ($F(3,36)=49.4$, $p<0.001$, multivariate results) and trial phase ($F(2,37)=75.9$, $p<0.001$). Bonferroni-corrected pairwise comparisons showed that conceptualizing actions as strategies was better than all other models (all $p<0.001$) but the other three models were not statistically different (Fig. 4f). Moreover, action decoding was highest at lever onset and lowest prior to cue onset (all Bonferroni-corrected pairwise comparisons with $p<0.001$).

Human data (new material added to Fig. 5, new Supplementary Fig. 9):

- We used a cluster-based permutation test (Maris E et al., *J Neurosci Methods* 2007) to show that decoding accuracy (decoding of attention focus) in Fig. 5i increases significantly after stimulus presentation (Fig. 5j).
- The spatial topography of this decoding result was presented at the source level in the main figure (was Fig. 5h, now Fig. 5k). We added a corresponding sensor-level analysis for the source-level results as requested by reviewer 1 (Supplementary Fig. 9b).
- A source-level fixed effects analysis was added to show that strategies can also be decoded from human MEG (Supplementary Fig. 9c, d).

3. The manuscript uses various forms of computational models without formally motivating them.

Following the reviewer's suggestion, we motivate the computational models in more detail in the revised manuscript (pp. 10-12). Briefly, we clarify that in the ACL model, by following a simple strategy, rats and humans only need to consider a 1-dimensional state space when making a choice (pressing right or left; attention at choice), and when assigning reward credit to that choice which drives learning (attention at learning). This is contrasted with having to deal with the full 4-dimensional state space of the task, modelled with the standard 4D-RL model (or simpler, but still higher dimensional 3D-RL and 2D-RL models). We clarify that by using attention to bias learning and choice in the ACL model, attention acts as a strong all-or-nothing filter that only allows one task feature (defined by the current strategy) to influence behavior.

4. Technical phrases, often used in the field in specific ways, are utilized in conjunction with each other without sufficient clarity, making the manuscript very hard to follow. Two examples are the following: strategy-specific selective attention, and value-based selection of behavioral strategies.

We made an effort to understand how different computational processes that are usually not considered together in the computational literature such as attentional processes and hierarchical cognition contribute to task rule inference. Technical phrases were thus used in conjunction to highlight connections between different computational layers but we should have made clearer our use of terminology. Thank you for alerting us to these potential sources of misunderstanding. We now went through the whole text and clearly defined such conjunctions on first usage. In our revised Discussion (pp. 27-31), we also discuss our findings in the context of different computational models and provide a new summary figure (Fig. 6).

For instance, by "strategy-specific selective attention" we refer to the current attention focus (definition now highlighted in legend of Fig. 2a). In computational terms, this refers to the attended-to sub-state space.

Similarly, by "value-based selection of behavioral strategies" we meant that not single stimulus-action pairs but whole task-sets are selected based on their reward value. In computational terms, this would be the action value of a task-set in a hierarchical RL model (Fig. 6). We do not model strategy selection but test behavioral (Fig. 3e-g) and neural predictions (Figs. 4, 5) made by such models.

5. Many figure panels have some visualization issues. For example, Fig.2 e,f and g the numbers are the x axis missing, making it hard to understand the figures.

Thank you for your comment. Sorry for this omission, we added trial numbers accordingly.

We agree that visualization is key to convey a complex story. We therefore went through all the figures to improve visualizations and made the following changes:

- We added an additional schematic of the task design in rat (Fig. 2a) and human experiments (Fig. 5a).
- We added a new schematic (Fig. 6) to summarize the computational processes underlying task rule.
- To make the distinction between task rules and behavioral strategies clearer, task rules are now presented with a rectangular shape and strategies with an oval-shaped icon (see Figs. 1b for an example).
- Learning curves: we replaced CUSUM performance plots and instead used sigmoidal model fits to visualize sudden transitions during learning. We also added learning curves based on sigmoidal model fits to Figs. 2e-g and 5f-h.
- Visualization of strategy sequences: All figures with a visualization of strategy sequences are now presented as in Fig. 2e-g to avoid visual artifacts (pointed out by reviewer 3, minor comment #3). We also added a visualization of the current task rule to all figures that display strategy sequences.
- We adjusted Fig. 2a to better visualize how attention organizes choice and learning during hypothesis testing.
- Complicated subfigures that are not central to the main thread of the paper are now presented in the Supplementary Information (previous Fig. 3h-j is now Supplementary Fig. 7b-d, previous Fig. 4e-g is now Supplementary Fig. 8).

REVIEWER COMMENTS

Thank you for the overall positive evaluation of our revision. We appreciate the many thoughtful comments that we address below using point-by-point responses.

Reviewer #1 (Remarks to the Author):

The authors have provided a careful response to the critiques of the four referees. In particular they have carefully addressed elements of the manuscript which were not entirely clear in the original submission. This has improved its accessibility. Thank you very much for the positive evaluation of our revision.

I still struggle with the human data. The authors have expanded on points of clarification sought in my original review. The broad claim, as stated in the abstract, is that neural substrates of hypothetical rule were detected in pre-frontal activity in both rodents and humans. This entails making a claim for species conserved mechanisms, where this refers to reduced task dimensionality. This begs a huge question as to the “homology” between the rodent and human task version. Likewise, the claim that the findings narrow the explanatory gap between human macroscopic and rodent microcircuit levels is quite a claim, and not entirely convincing based on the data.

So my remaining concern is with the exposition of the MEG data and the claim this is homologous to mouse data!

We made an effort to use aligned task versions in both species. More specifically, the human task version (Fig. 5a) uses the same stimulus material and corresponds to the paradigm where rats perform multiple, un-cued rule switches (Fig. 2a). At the same time, we aimed to both accommodate all rule switches in one MEG session and to avoid a ceiling effect. Based on a behavioral pilot outside of the scanner, we therefore decided to use probabilistic reward feedback in humans (as opposed to deterministic reward feedback in rats) and a fixed number of trials/rule (as opposed to performance-dependent rule switches in rats). Moreover, the human task version had an additional task rule with randomized reward feedback at the beginning to examine whether hypothesis testing reflects a general cognitive strategy, even when there is no behavioral advantage (similar to what we found in a cohort of naïve rats performing the random rule; Supplementary Fig. 3). In the first revision, we further used this data to conduct additional analyses in rats and humans (Figs. 1c, 5b) to show that behavior is better explained by the use

of low-dimensional as opposed to high-dimensional strategies or random choices. We now added a more detailed description of the human paradigm to the Methods section (pp. 40-41).

However, it is not sufficient to align task versions but it is crucial to show that cross-species assays also engage similar cognitive processes. Thus, our behavioral and computational analyses aimed to demonstrate that rats and humans indeed utilize similar computational mechanisms during rule learning. The results from both species converged to the same computational mechanisms, namely strategy-based learning and associated attentional mechanisms, which reduce task dimensionality and thereby promote cognitive flexibility (see summary Fig. 6). The main goal of our neural decoding analyses was to further provide evidence that such computational processes are actually implemented in similar brain regions. This is reminiscent of what has been found for other RL quantities such as action values or reward prediction errors in human fMRI or animal single-unit recordings (Schultz W et al., *Curr Opin Neurobiol* 2017; Lee D et al., *Annu Rev Neurosci* 2012). It is within this specific context that our study narrows the gap between findings based on human brain imaging and rodent cellular electrophysiology. In other words, neural decoding analyses provide important further evidence for our central claim that shared computational mechanisms exist in rats and humans. We clarified this in the Discussion (pp. 32-33).

In the Discussion, we also stress further limitations including additional human-specific cognitive processes during rule learning (Donoso M et al. *Science* 2014), that large parts of human PFC do not have clear corresponding homologues in rodents (Laubach M et al., *eNeuro* 2018), and that there is no one-to-one mapping between macroscopic imaging findings and electrophysiological measures at the microcircuit level (Nour MM et al., *Neuron* 2022).

Reviewer #2 (Remarks to the Author):

The authors have responded to all my comments with extensive revisions to the manuscript. I support publication of the manuscript.

We are glad to hear we could address all of your concerns. Thank you for your positive evaluation and your contributions to improving our manuscript.

Reviewer #3 (Remarks to the Author):

Species-conserved mechanisms of abstract rule learning promote cognitive flexibility in complex environments

Bähner, Popov, Boehme, Hermann, Merten, Zingone, Koppe, Meyer-Lindenberg, Toutounji, and Durstewitz

This is a re-review of this manuscript. As before, I feel this is an interesting manuscript that uses a series of complex behavioral and neural analyses to study strategy switching in both rats and humans. In my opinion, the manuscript is timely and will be of broad interest. The revisions strengthened an already strong manuscript and largely addressed my previous comments. I clarified a couple of our previous comments below, which I hope the authors would be able to address.

Thank you for the encouraging assessment of our revision. Below we address all remaining points.

1. Our previous comment #4 noted that it wasn't clear to us how the ACL model differed from attention to sensory inputs. I apologize if our comment wasn't clearly communicated, but I'm not sure the authors response address this concern. In their response, the authors seem to first agree that the ACL model is equivalent to a sensory filtering model. But then they appear to address a slightly different question – whether attention is either binary or graded. Either binary or graded attention could be encapsulated by sensory filtering (I think the reviewers agree with this as well?). To restate the previous point – the literature tends to discuss attention as acting on sensory inputs. Filtering at an early stage would impact both choices and learning. Here, the equivalent model seems to be the ACL model where attention acts on both choice and learning. As noted before, these seem equivalent and I would suggest a) making this clear in the manuscript and b) adopting the simpler terminology/framework used in previous work, where attention modulates stimulus inputs. I would strongly recommend the authors at least do (a) but would leave (b) up to them and their style.

Thank you for coming back to this issue. Yes, we agree that the ACL model is equivalent to sensory filtering models and that these models encompass both binary and graded filters. We compared the binary vs. graded ACL model to provide evidence for strong dimensionality reduction during strategy-based learning. However, we agree that our use of terminology can be misleading given the rich literature on sensory filtering in selective attention (e.g., Driver J,

Br J Psychol 2001; Treisman AM, *Psychol Rev* 1969). We now made this connection between ACL and the sensory filtering model more explicit in the Discussion (p. 31). Given that our modelling approach is an adaptation of attention-based RL models as in Leong YC et al. (*Neuron* 2017), we decided to keep the terminology as introduced in that work, however.

2. Our previous comment #8 noted that, given the fact that the animals orientation and location in space was correlated with their behavioral strategy, the neural responses observed in prefrontal cortex of rats could reflect those behavioral variables rather than (or in addition to) encoding the behavioral strategy. In their response the authors address a potential motor confound. Does this also encapsulate the position differences (i.e., without the motor action itself?). Are head direction, body orientation, and location all perfectly correlated with the motor response? My impression from the manuscript was that this was not true, which means there could still be a correlation between orientation/location and strategy even after controlling for motor responses. In this case, our question remains – can these be fully dissociated? If not, I would suggest noting this potential confound in the Discussion.

Thank you for this important comment. On the one hand, Fig. 4g, h shows that PFC population activity represents abstract strategies rather than simple motor actions. Moreover, strategies could be decoded throughout all trial phases, including the inter-trial interval. Rats were freely moving in the operant chamber and their orientation and location in space was therefore highly variable both within and across trials. We therefore find it highly unlikely that strategy decoding is simply a reflection of movement parameters. On the other hand, we provided evidence for strategy-specific movement patterns that included both head direction plots which correspond to the strategy-specific attention focus (Fig. 2b, Supplementary Figs. 2-4) and also movement patterns during the inter-trial interval (Supplementary Fig. 5). Moreover, attention-at-choice indeed affected prefrontal strategy decoding (Fig. 4d). In this case, our behavioral read-out for attention is the orientation reaction following cue onset and it is therefore not possible to disentangle movement from cognition completely. We clarified this in the Discussion section (p. 33).

3. Our previous comment #3 was asking whether removing strategies from the model fits reduced the accuracy of the overall fit – that is, were the animals always using all possible strategies or could some of them be removed from the fit? The authors response was not clear – I understand that the rat would eventually be trained on tasks that required the use of all

strategies, but this doesn't mean that the animal was always using or sampling all of the strategies throughout the process. Similarly, one could imagine adding new strategies to the mix, even if they were never tested in a task, and improving the model's fit to the animals behavior (this was the intention behind what the authors designated as comment 3c; we incorrectly wrote 'models' instead of 'strategies' which probably led to the confusion).

We are sorry for the misunderstanding. We think that two points can be made to address this issue.

First, it is correct that rats were not always sampling all strategies during learning. Across all task rules, animals followed a median of 7 (6-8) out of 8 possible strategies (Results p. 6). The same is true for humans (6 (5-6) out of 8 strategies, Results p. 25). We therefore now compared the binary ACL model to three model variants where one of the state-action-spaces (either auditory, visual or place-outcome) was treated as in the UA (uniform attention) model. This corresponds to a scenario in which strategy-specific attention effects for either auditory (*go-click, go-silent*), visual (*go-light, go-dark*) or place-outcome strategies (*alternate, go-left, go-right and win-stay-lose-shift*) are removed. Both in rats and humans performing multiple rule switches, the ACL model was better at predicting held-out behavioral responses as compared to all three model variants (new Supplementary Fig. 6d, e).

Second, RL modelling of higher-dimensional strategies would require larger state-action spaces because these strategies all depend on two task features (e.g., for *win stay-lose shift visual* the decision to press the right vs. left lever depends on both the current cue light location and the previous reward). Such RL models were not included because in previous analyses we found no experimental evidence for high-dimensional strategies in either rats or humans (Figs. 1c, 5b).

We included these additional model findings in the respective Results sections (p. 13, p. 26).

Reviewer #4 (Remarks to the Author):

The manuscript has improved in presenting the behavioral data.

We are glad to hear we could address your concerns with respect to behavioral data.

The rat mPFC results are still not convincing. I appreciate that authors have attempted to show less processed neuronal data. However, what is shown as an 'example neuron' is arguably an outlier, as evident in the histogram of fig 4c. Moreover, it remains unclear how neurons respond to task events. In past studies a significant proportion of mPFC neurons respond to events (cues, lever presses) in a task with transient changes in the firing rate, and this does not happen to be the case here, as shown in fig 4c. Given this, the quality of spiking data is unclear to me, and the example neuron shown suggest to me that the recordings are not sufficiently stable.

We are sorry that we created the false impression that only few neurons are task-responsive in our experiments. It is correct that the PFC literature indicates low signal-to-noise ratio (SNR) for individual units in contrast to high SNR at the population level (Durstewitz D et al., *Neuron* 2010; Rigotti M et al. *Nature* 2013; Tye KM et al., *Neuron* 2024). However, this does not imply that individual PFC units are not task-responsive. The references we cite in this manuscript report that the percentage of PFC units that significantly discriminate between task rules was >30% in rats (Durstewitz D et al., *Neuron* 2010) and either ~33% (Mansouri FA et al., *J Neurosci* 2006) or 41% (Wallis JD et al., *Nature* 2001) in primates. Moreover, many prefrontal units are mixed selectivity neurons, which means that they are responsive to multiple, and not only a single task feature (Rigotti M et al. *Nature* 2013; Tye KM et al., *Neuron* 2024).

We now tested whether single-unit firing significantly discriminated between either different strategies, side of lever press, reward feedback or location of visual and auditory cues (Bonferroni-corrected unpaired t-tests with $p < 0.05$). According to these criteria, 67.8% of all units were responsive to at least one task feature. The firing rate of 37.1% of units discriminated between at least one pair of strategies in at least one task phase. Unit firing was also significantly modulated by the side of lever press (40.9%), reward feedback (28.9%) or sensory cue location (visual cue: 11.9%, auditory cue: 7.6%). Moreover, many units were responsive to multiple task features (36.8 of units responded to $\geq 2/5$, 15.1% to $\geq 3/5$ of the tested task features; see example units in Fig.4c and Supplementary Fig. 8a-c). Thus, our findings are fully consistent with the cited literature on single-unit selectivity in PFC.

We further computed the selectivity index d' for strategies during three trial phases for 3348 strategy pairs from 1884 units: 0.2 (0.1-0.34) pre cue, 0.21 (0.1-0.36) post cue and 0.22 (0.1-0.39) post lever presentation (see distribution of d' values in Supplementary Fig. 8d) which is similar to d' for task rules (see Fig. 1 in Durstewitz D et al., *Neuron* 2010). This is in line with a low SNR at the single-unit level because $d' > 2$ is required to reach a misclassification rate of

less than 8% based on normality assumptions (Durstewitz D et al., *Neuron* 2010) and $d' > 2$ was only observed in 3/1884 units.

The example neuron you are referring to was picked to illustrate that above-chance decoding prior to cue onset also exists at the single-neuron level (46.4% of strategy-selective units significantly discriminated between at least one pair of strategies before cue onset). However, it is correct that this unit is an outlier based on the definition of a value greater than $Q_3 + 1.5 \times$ interquartile range/IQR (see Methods). This corresponds to $d'_{\text{pre cue}} > 0.7$, $d'_{\text{post cue}} > 0.75$ and $d'_{\text{post lever}} > 0.81$. We therefore replaced the unit shown in Fig. 4c and show further examples (new Supplementary Fig. 8a-c) of units that significantly discriminate between strategy pairs but are not outliers as defined above.

We now also performed two further analyses to show that high SNR at the population level depends on small contributions from multiple, strategy-selective units. First, we compared population level decoding based on all units in a session with accuracy after sequentially removing one unit at a time. On average, the resulting difference in decoding accuracy per unit was small in all trial phases: -0.27 ($-1.4 - +0.87$) % pre cue, -0.27 ($-1.53 - +0.93$) % post cue and -0.2 ($-1.27 - +0.8$) % following lever onset (negative values indicate a decrease in decoding accuracy after neuron removal; Supplementary Fig. 8e). Linear regression (Supplementary Fig. 8f) confirmed that accuracy differences were predicted by the respective selectivity index for strategy in all three trial phases ($p_{\text{pre cue}} = 4.2 \times 10^{-117}$, $p_{\text{post cue}} = 2 \times 10^{-106}$, $p_{\text{post lever}} = 2.8 \times 10^{-105}$) but other regressors in the model were not consistent predictors. More specifically, we excluded that neurons with strong firing rate changes across trials (measured as the unit's maximum firing rate change associated with a neural change point, see Supplementary Methods for details) influence strategy decoding (all $p > 0.05$). Moreover, neither the total neuron number recorded in a session ($p_{\text{pre cue}} = 0.74$, $p_{\text{post cue}} = 3.4 \times 10^{-3}$, $p_{\text{post lever}} = 0.06$) nor the number of task features a unit coded for ($p_{\text{pre cue}} = 0.02$, $p_{\text{post cue}} = 0.94$, $p_{\text{post lever}} = 0.55$) were consistent predictors. Second, we ranked all units in a session according to d' strategy in each trial phase and repeated population decoding after removing the top 5, 10, 20, 30 or 40% units from the strategy decoding analysis. Only after removing the top 40% of units, decoding accuracy was not higher than 50% chance level (one-sample t-test with $p > 0.05$) and this effect was restricted to the respective trial phase (Supplementary Fig. 8g). Results from the analyses above are described on pp. 19-20.

Figures have improved given my previous comments. However, some labels are still missing.

We carefully checked all figures again and made the following corrections:

- Figure 3a, right panel: a unit was missing on the y-axis - [trials]
- Supplementary Fig. S5: we added a unit to all y-axes - [a.u.]

REVIEWER COMMENTS

Thank you for the overall positive evaluation of our second revision. We appreciate your thoughtful comments that we address below using point-by-point responses.

Reviewer #1 (Remarks to the Author):

My concerns related to homology of MEG and rodent task remain. I still consider that the linkage is weak notwithstanding the abbreviated presentation of the MEG findings! At the very least I would strongly advocate that the manuscript title is changed - given the controversial issue around this particular issue

We agree that it is difficult to define what constitutes a species-conserved mechanism and therefore removed the species conservation element from the title as requested. At the same time, this study uses multiple analyses to characterize similar learning mechanisms in both species and we therefore propose a title that reflects the translational perspective but makes a weaker statement regarding homology of the findings: “Abstract rule learning promotes cognitive flexibility in complex environments across species”.

To further strengthen the analysis - particularly the MEG findings that were the focus of critique - and to support future replication efforts, we have added a modified, more comprehensive topographical visualization of the main results in Fig. 5k, in addition to the details that had been already provided in Fig. 5 and Supplementary Fig. 10. More specifically, we computed a within-subject contrast (paired t-test) between the task condition (0 to 1000 ms following cue presentation) and baseline (-500 to 0 ms) and present a detailed map of Cohen’s d effect sizes for this key contrast. This not only enhances the interpretability and transparency of the results but also provides a basis for future power analyses and sample size estimations, thus contributing to cumulative and reproducible progress in evaluating the degree to which the proposed mechanisms are shared across species. The resulting spatial distribution of Cohen’s d aligns with expectations based on task demands, showing robust effects in primary auditory and visual cortices as well as the prefrontal cortex — areas known to be engaged in sensory processing and executive control.

Reviewer #3 (Remarks to the Author):

The authors have addressed my comments. Congratulations on a very interesting paper!

We are glad to hear we could address all of your concerns. Thank you for your positive evaluation and your contributions to improving this manuscript.